# MOSAiC studies of long-lasting mixed-phase cloud events and analysis of the liquid-phase properties of Arctic clouds

Cristofer Jimenez<sup>1</sup>, Albert Ansmann<sup>1</sup>, Kevin Ohneiser<sup>1</sup>, Hannes Griesche<sup>1</sup>, Ronny Engelmann<sup>1</sup>, Martin Radenz<sup>1</sup>, Julian Hofer<sup>1</sup>, Dietrich Althausen<sup>1</sup>, Daniel A. Knopf<sup>2</sup>, Sandro Dahlke<sup>3</sup>, Johannes Bühl<sup>1,4</sup>, Holger Baars<sup>1</sup>, Patric Seifert<sup>1</sup>, and Ulla Wandinger<sup>1</sup>

Correspondence: C. Jimenez

(jimenez@tropos.de)

#### Abstract.

Vertically resolved observations of the temporal evolution of mixed-phase clouds (MPCs) were performed over the central Arctic during the MOSAiC (Multidisciplinary drifting Observatory for the Study of Arctic Climate) expedition which lasted from October 2019 to September 2020. The research icebreaker *Polarstern*, drifting with the pack ice for more than seven months, mostly at latitudes >85°N, served as a platform for state-of-the-art remote sensing of aerosols and clouds. The use of the recently introduced dual field-of-view (FOV) polarization lidar technique in combination with the well-established lidarradar retrieval technique provided, for the first time, a robust instrumental basis to monitor the evolution of the liquid and the ice phase of MPCs and the interplay between the two phases. Two long-lasting Arctic MPC events observed close to the North Pole in mid winter (December 2019) and late summer (September 2020) are discussed to provide new insight into Arctic MPC evolution processes. In the second part of the article, cloud statistics, covering all seasons of a year, are presented. The focus is on the optical and microphysical properties of the liquid phase. These results are solely derived from the dual FOV lidar observations. The key findings of the study can be summarized as follows: Persistent activation of aerosol particles to form water droplets is of great importance for the longevity of MPCs. The observations confirm that ice formation occurs predominantly via immersion freezing. The field studies suggest that the free tropospheric reservoirs of cloud condensation nuclei (CCN) as well as of ice-nucleating particles (INPs) were always well filled, i.e., the clouds did not exhaust their supply of activatable and activated particles. The observation of long-lasting MPC events, low ice production rates, and a sufficiently large INP reservoir lead to the recommendation to use a time-dependent immersion freezing parameterisation in MPC modeling efforts.

## 1 Introduction

Stratiform mixed-phase clouds (MPCs) occur everywhere around the globe from the tropics to the poles, at all altitudes with temperatures from  $0^{\circ}$  down to  $-36^{\circ}$ C. Arctic MPCs exert a sensitive impact on the radiation field and temperature conditions,

<sup>&</sup>lt;sup>1</sup>Leibniz Institute for Tropospheric Research, Leipzig, Germany

<sup>&</sup>lt;sup>2</sup>School of Marine and Atmospheric Sciences, Stony Brook University, Stony Brook, NY 11794, USA

<sup>&</sup>lt;sup>3</sup>Alfred Wegener Institute, Helmholtz Centre for Polar and Marine Research, Potsdam, Germany

<sup>&</sup>lt;sup>4</sup>Harz University of Applied Sciences, Wernigerode, Germany

on the evolution of precipitation, and thus on the vertical exchange of water in the polar troposphere (Lohmann and Neubauer, 2018; Wendisch et al., 2019; Morrison et al., 2020). Therefore, a careful consideration of these MPCs in climate and weather prediction models is a clear need. However, because these clouds show complex microphysical and thermodynamic properties, have life times ranging from minutes to days, and their evolution is influenced by many atmospheric (meteorological) processes and varying environmental (aerosol) conditions, they are difficult to model and to be appropriately parameterized in atmospheric simulation models (Savre and Ekman, 2015; Solomon et al., 2018; Fridlind and Ackerman, 2018).

More life cycle studies are needed to accelerate progress in this important field of atmospheric research. The life cycle of a stratiform MPCs is controlled by two fundamental processes. The formation and maintenance of a droplet-dominated shallow cloud layer at the MPC top is of key importance. This steers all further processes that are required to explain the long MPCs lifetimes often observed in the Arctic. Emission of infrared radiation by the water droplets causes strong cooling at cloud top which leads to negative buoyancy at the stable conditions and subsequently to the evolution of a complex field of upward and downward motions below and within the liquid-dominated cloud layer (Shupe et al., 2008; Roesler et al., 2017). In areas with updrafts, existing droplets can grow and new droplets can form at water-supersaturation conditions. The steady resupply of water droplets allows the MPC supercooled liquid layer to persist despite the continuously forming ice. In the Arctic, lifetimes can be many hours or even days (Morrison et al., 2012).

The second fundamental process is heterogeneous ice nucleation in the liquid-dominated cloud layer, the subsequent growth of the ice particles, and the formation of extended ice virgae of falling ice crystals (Rauber and Tokay, 1991; Shupe et al., 2008; Ansmann et al., 2009; Morrison et al., 2012). The evolution of virgae with growing and sublimating ice crystals in different parts of the fall streaks influence cloud dynamics and lifetime (Korolev and Field, 2008; Morrison et al., 2011; Eirund et al., 2019). Immersion freezing, i.e., ice nucleation on an ice-nucleating particle (INP) inside a supercooled water droplet, is the main ice nucleation mode (Ansmann et al., 2008, 2009), and preferably starts in the coldest region of the liquid cloud layer, i.e., at cloud top (Hobbs and Rangno, 1985), and preferably during updraft periods (Shupe et al., 2008; Ansmann et al., 2009). We would like to emphasize here that the first clear hint for the dominance of immersion freezing (compared to other ice nucleation modes) in stratiform MPCs was reported for subtropical and tropical MPCs (Ansmann et al., 2009), based on lidar observations during the SAMUM (Saharan Mineral Dust Experiment) campaigns (Ansmann et al., 2011). Motivated by the subtropical and tropical studies, this finding was confirmed for mid-latitudinal MPCs (Westbrook and Illingworth, 2011) and Arctic stratiform clouds (de Boer et al., 2011). Rauber and Tokay (1991) and Shupe et al. (2008) already hypothesized that the liquid phase is obviously needed to trigger ice nucleation. Meanwhile, immersion freezing is well recognized as the main ice nucleation mode in mixed-phase clouds (Boucher et al., 2013; Vali and Snider, 2015; Savre and Ekman, 2015; Fridlind and Ackerman, 2018; Solomon et al., 2018; Khain et al., 2022; Knopf et al., 2023). Further evidence for the predominance of the immersion freezing mode will be given in this article.

An important aspect, presently in the discussion, is the parameterization of immersion freezing in atmospheric models. Two different approaches are available, the time-independent (diagnostic) approach and the time-dependent (prognostic) approach (Knopf et al., 2023). The time-dependent approach to immersion freezing is following the classical nucleation theory (CNT) (Knopf and Alpert, 2013; Alpert and Knopf, 2016; Knopf et al., 2020). The time-independent approach includes a

time-independent particle-number- and surface-area-based descriptions of ice nucleation (DeMott et al., 2015; Ullrich et al., 2017). In the latter approach, INPs are unique among a population of same aerosol particles, e.g., dust particles. Only a small fraction of the particles can serve as INPs, i.e., belong to the activatable particle fraction. In the time-dependent approach, all particles can be activated at a random base and belong to the reservoir of INPs within a given cloud layer. Thus, the choice of the parameterization defines the size of the INP reservoir. The CNT-based time-dependent description yields an orders of magnitude larger INP reservoir than the alternative, time-independent parameterization. A sufficiently large INP reservoir is required to explain the longevity of Arctic MPC systems (Knopf et al., 2023). We will further discuss this INP reservoir aspect in Sect. 4.1 and 4.2.

This study has two goals. In the first part, we discuss two long-lasting MPC events (one winter and one late summer case) and provide new insight into the evolution of Arctic MPCs. The observations were performed in the framework of the MOSAiC (Multidisciplinary drifting Observatory for the Study of Arctic Climate) 2019-2020 expedition (Shupe et al., 2022). The German research icebreaker *Polarstern*, drifting with the pack ice for more than seven months, from October 2019 until mid-May 2020, mostly at latitudes >85°N, served as a platform for state-of-the-art remote sensing of aerosols and clouds. The use of the recently introduced dual field-of-view (FOV) polarization lidar technique (Jimenez et al., 2020a, b) in combination with the well-established lidar-radar retrieval technique provided, for the first time, a robust instrumental basis to monitor the evolution of the liquid and the ice phase of MPCs separately and the interplay between the two phases.

65

80

The liquid phase properties such as the droplet effective radius  $R_{\rm e,liq}$ , cloud light-extinction coefficient  $\alpha_{\rm liq}$ , liquid water content (LWC), and cloud droplet number concentration (CDNC) were derived from the dual FOV lidar measurements. This is an important new contribution to experimental Arctic cloud research. We implemented the new lidar technique into a state-of-the-art cloud-aerosol Raman lidar just two months before the start of the MOSAiC expedition. The corresponding microphysical properties of the ice crystals were obtained by combing the lidar backscatter observations at 532 nm wavelength and cloud radar reflectivity measurements at 8.5 mm wavelength, measured in the ice virgae just below the liquid-water-containing cloud layer (Bühl et al., 2019a; Ansmann et al., 2019b, 2025a).

The MOSAiC observations allowed us to study the evolution of MPCs in an orographically undisturbed environment with homogeneous surface characteristics so that cloud evolution was a function of meteorological and aerosol conditions, only. The cloud layers developed in Arctic haze during the winter half year. In the few summer months, cloud evolution occurred in a complex mixture of aged anthropogenic hazes, soil and desert dust, and wildfire smoke, transported from remote continents (Ansmann et al., 2023). Local marine particles and particles of biogenic origin contributed to the aerosol composition in the lower troposphere as well.

In the second part of the article, the results of the statistical analysis of the liquid-phase microphysical properties of liquid-containing cloud layers, observed from October 2019 to September 2020, are presented and discussed. The corresponding statistics for the ice phase can unfortunately not be provided. The statistical analysis of the large MOSAiC remote sensing data sets requires automated versions of the retrieval procedures. Such an automated version was developed in the case of the dual-FOV-lidar data analysis scheme, but could not be realized until now in the case of the lidar-radar retrieval procedure. The

lidar-radar data analysis is complex and time consuming and includes a careful selection and setting of input parameters. As a consequence, this retrieval scheme could only be applied to a few case studies.

The article is organized as follows: After an introduction to the MOSAiC *Polarstern* route, employed instruments, and applied data analysis methods in Sect. 2, a brief overview about the aerosol conditions in the free troposphere is given in Sect. 3, followed by a discussion of the temporal evolution of two long-lasting MPC events in Sect. 4. The statistical results of Arctic MPC properties are presented in Sect. 5. Finally, in Sect. 6, we briefly provide some comments regarding the conceptual MPC life cycle model that covers all key processes and aspects of an MPC life cycle and was introduced by Shupe et al. (2008) and Morrison et al. (2012). We provide some kind of an update gained from our MOSAiC observations. Concluding remarks are given in Sect. 7.

## 100 2 MOSAiC instrumentation and observational products

## 2.1 Remote sensing aboard Polarstern

105

110

MOSAiC was the largest Arctic research initiative in history. The main measurement period lasted from the beginning of October 2019 until the end of September 2020. The goal of the MOSAiC expedition was to take the closest look ever at the Arctic as the epicenter of global warming and to gain fundamental insights that are key to better understand global climate change. A detailed monitoring of the atmosphere, the ocean, the cryosphere and biosphere in the Central Arctic was realized. Most of the MOSAiC observations were conducted at latitudes >85°N (from the beginning of October 2019 to the beginning of April 2020, and from mid-August to the end of September 2020). Observations during the summer season (mid-May to mid-August) were performed at latitudes from 79-82°N, north and northwest of Svalbard, Norway.

Our role in the MOSAiC consortium was to provide a seasonally and height-resolved characterization of aerosols and clouds in the North Pole region from the surface up to 30 km height (Engelmann et al., 2021; Ohneiser et al., 2023; Ansmann et al., 2023, 2024, 2025a, b; Griesche et al., 2024b). The German icebreaker *Polarstern* (Knust, 2017) served as the main MOSAiC platform for advanced remote sensing studies of the atmosphere (Shupe et al., 2022). *Polarstern* was trapped in the ice and drifted with the pack ice through the Arctic Ocean from 4 October 2019 to 16 May 2020. The entire cruise of the *Polarstern* is shown in Shupe et al. (2022). Our state-of-the-art combined aerosol-cloud Raman and dual FOV polarization lidar (Engelmann et al., 2016; Jimenez et al., 2020b) aboard *Polarstern* was operated side by side with the ARM (Atmospheric Radiation Measurement) mobile facility 1 (AMF-1), which conducted cloud radar observations. Several MPC studies using MOSAiC remote sensing data are meanwhile published (Silber and Shupe, 2022; Saavedra Garfias et al., 2023; Barrientos-Velasco et al., 2025).

Ground-based active remote sensing with advanced lidars and cloud radars permits a detailed monitoring of the different stages of the life cycle of stratiform MPC systems with high vertical and temporal resolution over hours to days (Shupe, 2007; Shupe et al., 2008; Illingworth et al., 2007; de Boer et al., 2011; Bühl et al., 2016; Kalesse et al., 2016; Achtert et al., 2020; Griesche et al., 2020; Radenz et al., 2021b; Engelmann et al., 2021). Sophisticated retrieval methods (e.g., Delanoë and Hogan, 2008; Schmidt et al., 2013; Sourdeval et al., 2018; Bühl et al., 2019a; Jimenez et al., 2020a; Mason et al., 2023)

allow for a detailed derivation of cloud microphysical properties. Meanwhile, also lidar-based aerosol retrieval methods are introduced to cover the aerosol part in cloud process studies (Mamouri and Ansmann, 2016; Marinou et al., 2019; Ansmann et al., 2019a, 2021; Choudhury and Tesche, 2022; Choudhury et al., 2022; He et al., 2023). The dual FOV Raman lidar technique (Schmidt et al., 2013) and the new dual FOV polarization lidar technique (Jimenez et al., 2020a) were developed for an improved monitoring of life cycles of stratiform clouds with focus on the liquid phase and for an improved observation-based study of the impact of aerosols on the microphysical properties of stratiform, droplet-dominated cloud layers (Schmidt et al., 2014, 2015; Jimenez et al., 2020b).

Besides the lidar observations we made use of the MOSAiC ARM cloud radar measurements to derive profiles of ice crystal properties of the MPC systems by combining lidar and radar data. The 35 GHz Doppler cloud radar (Ka-band ARM Zenith Radar, KAZR) of the ARM mobile facility AMF-1 of the US Department of Energy (http://www.arm.gov, last access: 22 November 2024) measures profiles of radar reflectivity, mean Doppler velocity, and Doppler spectrum width (ARM, 2024).

We also used the retrievals of the liquid water path (LWP) with the radiometer HATPRO (Humidity and Temperature Profiler) microwave radiometer (Rose et al., 2005; Griesche et al., 2024b) mounted on the roof of our lidar container. Finally, we used the dense set of radiosonde temperature and relative humidity profiles (Maturilli et al., 2021) in our cloud studies. Vaisala radiosondes (type RS41) were launched regularly every 6 hours aboard *Polarstern* throughout the entire MOSAiC year.

## 2.2 Raman lidar products



An overview of the aerosol and cloud products, used in this study and obtained from the lidar and radar observations are listed in Table 1. The errors given in Table 1 can be regarded as typical uncertainties and were partly obtained from validation efforts (Ansmann et al., 2023). The basic lidar data analysis applied to obtain the geometrical (cloud base and top heights) and optical (backscatter, extinction, linear depolarization ratio) properties is outlined in Baars et al. (2016), Haarig et al. (2016), and Hofer et al. (2017). The main features of the MOSAiC-related aerosol data analysis (including signal correction, Rayleigh backscattering and extinction correction, temporal averaging and vertical smoothing of signal profiles) are described in Ohneiser et al. (2020, 2021, 2022). Relative-humidity fields, as shown in Engelmann et al. (2021), Seidel et al. (2024), and in Sect. 4 are obtained from Raman lidar observations of the water-vapor-to-dry-air mixing-ratio profiles (Dai et al., 2018) and temperature profiles measured with the *Polarstern* radiosondes. In this computation, the temperature profiles are required with a resolution of 30 s and obtained by linear interpolation between the radiosonde data for each height bin, given for fixed times (6, 12, 18, and 24 UTC). Quality checks were based on comparisons of the Raman-lidar with the respective MOSAiC radiosonde humidity profiles (Seidel et al., 2024).

## 2.3 POLIPHON: CCN and INP reservoirs

The retrieval of aerosol microphysical properties such as the number concentrations  $n_{50,\text{dry}}$  and  $n_{250,\text{dry}}$  of dry aerosol particles, listed in Table 1, is performed by means of the POLIPHON (Polarization Lidar Photometer Networking) approach (Mamouri and Ansmann, 2016, 2017; Ansmann et al., 2023). The particle number concentration  $n_{50,\text{dry}}$ , considering all dry particles with radius >50 nm, is used as proxy for the reservoir of cloud condensation nuclei (CCN). The number concentration

**Table 1.** Aerosol and cloud products, obtained from lidar-only analysis (lidar) or from the synergy of lidar and radar observations (LIRAS: lidar-radar synergy). r denotes the radius of the dry aerosol particles. Typical relative uncertainties in the retrieval products are given. The referenced articles provide detailed information about the retrievals and further literature.

| Aerosol and cloud properties                                         |                        | Method | Uncertainty   | Reference                 |
|----------------------------------------------------------------------|------------------------|--------|---------------|---------------------------|
| Aerosol particle number conc. $(r > 50 \text{ nm}) [\text{cm}^{-3}]$ | $n_{50,\mathrm{dry}}$  | Lidar  | 50-75%        | Ansmann et al. (2023)     |
| Aerosol particle number conc. ( $r>250~\mathrm{nm}$ ) [cm $^{-3}$ ]  | $n_{250,\mathrm{dry}}$ | Lidar  | 25-50%        | Ansmann et al. (2023)     |
| Cloud droplet extinction coefficient [Mm <sup>-1</sup> ]             | $lpha_{ m liq}$        | Lidar  | 15-20%        | Jimenez et al. (2020a, b) |
| Cloud droplet depolarization ratio                                   | $\delta_{ m v}$        | Lidar  | ≤5%           | Jimenez et al. (2020a, b) |
| Cloud droplet number concentration $[cm^{-3}]$                       | CDNC                   | Lidar  | 25-75%        | Jimenez et al. (2020a, b) |
| Cloud droplet effective radius $[\mu m]$                             | $R_{ m e,liq}$         | Lidar  | 15%           | Jimenez et al. (2020a, b) |
| Liquid water concentration $[\mu g \text{ m}^{-3}]$                  | LWC                    | Lidar  | 25%           | Jimenez et al. (2020a, b) |
| Ice crystal extinction coefficient [Mm <sup>-1</sup> ]               | $lpha_{ m ice}$        | Lidar  | ≤20%          | Haarig et al. (2016)      |
| Ice crystal number concentration $[L^{-1}]$                          | ICNC                   | LIRAS  | factor of 2-3 | Ansmann et al. (2025a)    |
| Ice crystal effective radius $[\mu m]$                               | $R_{ m e,ice}$         | LIRAS  | 35%           | Ansmann et al. (2025a)    |
| Ice water concentration [ $\mu$ g m <sup>-3</sup> ]                  | IWC                    | LIRAS  | 40%           | Ansmann et al. (2025a)    |

 $n_{250,\mathrm{dry}}$  considers the larger dry particles with radius >250 nm. According to DeMott et al. (2015), dust particles with radius > 250 nm are the most favorable INPs in the free troposphere. Therefore, the estimated dust particle number concentration  $n_{250,\mathrm{d.dry}}$  is used as a proxy for the INP reservoir in our study.



The required extinction-to-number-concentration conversion factors, needed to convert particle extinction coefficient profiles into  $n_{50,\rm dry}$  and  $n_{250,\rm dry}$  profiles, are derived from Arctic AERONET (Aerosol Robotic Network) sun photometer observation during the summer half year (Ansmann et al., 2023). For the winter half year, without sun light, conversion factors for the dominating Arctic haze aerosol type are not available. Fortunately, an episode with a strong Arctic haze event could be measured with our AERONET (Aerosol Robotic Network) sun photometer over Leipzig, Germany, in April 2002 (Müller et al., 2004). The obtained Arctic haze conversion factors were rather close to the summer Arctic aerosol conversion factors given in Ansmann et al. (2023). This finding encouraged us to apply the Arctic summer conversion factors to the entire MOSAiC data set to obtain year-round time series of  $n_{50,\rm dry}$  and  $n_{250,\rm dry}$  profiles. The multiplication of a given lidar-derived extinction coefficient (in Mm<sup>-1</sup>) with the conversion factor of 10 Mm cm<sup>-3</sup> yields the  $n_{50,\rm dry}$  value. In a similar way, multiplication with the conversion factor of 0.13 Mm cm<sup>-3</sup> leads to the  $n_{250,\rm dry}$  value (Ansmann et al., 2023). We obtain a realistic dust-related particle number concentration  $n_{250,\rm d,dry}$  by applying the conversion factor of 0.13 Mm cm<sup>-3</sup> and assuming a dust fraction of 1-10%, i.e.,  $n_{250,\rm d,dry}/n_{250,\rm dry}$  from 0.01 to 0.1. The assumption of a dust fraction of 1-10% in the free troposphere is supported by global airborne aerosol observations presented by Froyd et al. (2022).

## 2.4 Dual FOV lidar: optical and microphysical properties of water droplets








In this article, the focus is on the one-year MOSAiC observations of stratiform clouds in the lower and middle free troposphere at heights above 500 m. The Arctic boundary layer over Polarstern usually reached to heights below 500 m height Peng et al. (2023). The dual FOV polarization lidar method (Jimenez et al., 2020a, b), applied to gain the liquid droplet properties, is based on the measurement of the volume linear depolarization ratio  $\delta_v$  at two different FOVs. The depolarization ratio is defined as the ratio of the cross-polarized to the co-polarized backscatter coefficient, "cross" and "co" indicate the plane of polarization orthogonal and parallel to the plane of linear polarization of the transmitted laser pulses. The volume depolarization ratio monotonically increases from zero at cloud base to values > 0.2 within the liquid-dominated cloud layer due to multiple scattering of laser light by the cloud droplets. By measuring the multiple scattering effect (via light depolarization) with two FOVs we are able to unambiguously derive the single-scattering cloud extinction coefficient  $\alpha_{liq}$  and the effective radius  $R_{\rm e,lig}$ , i.e., the geometrical-cross-section-weighted mean radius of the droplets. The derived extinction coefficients can be cross-checked by the measured single-scattering droplet backscatter coefficients (multiplied by the water cloud lidar ratio of 18 sr) obtained from elastic backscatter signals. In the next step, we calculate the liquid water content, LWC, as a function of the cloud extinction coefficient and effective radius and, finally, the cloud droplet number concentration, CDNC, from LWC by assuming a gamma size distribution. The most accurate set of solutions is obtained at heights of 50-100 m above the base of the liquid-dominated cloud layer of the MPC (Jimenez et al., 2020b; Engelmann et al., 2021). In Fig. 1, the center height of 75 m above cloud base is highlighted as thick horizontal blue lines. To avoid a potentially sensitive bias in the retrieval products by the incomplete overlap between the laser beam and the different receiver FOVs of the lidar, we excluded the dual-FOV-lidar observations in the near range (<500 m height) from the further analysis. The lidar overlap profiles can always slightly vary, e.g., with changing temperatures in the air-conditioned lidar container. The lidar signal profiles are stored with 30 s resolution and averaged over seven minutes before the dual-FOV-lidar data analysis is applied.

In situ observations (Mioche et al., 2017) indicate that such observations in the lower part of the liquid-dominated cloud layer represent the liquid phase properties of the entire MPC up to cloud top well. The uncertainty analysis shows that the lidar approach, originally developed for pure water clouds, can be applied even to MPCs as long as the contribution of backscattering by ice crystals to the total backscatter coefficient is clearly below 5% in the liquid-bearing cloud layer (Engelmann et al., 2021). During MOSAiC, the ice backscatter fraction was always on the order of 1-2% or less at 50-100 m above cloud base. It should be mentioned that lidars are able to correctly measure the cloud optical depth up to around 2.5. This means that useful backscatter signals of a liquid-dominated cloud layer are available up to 100 m above cloud base in the case of cloud extinction coefficients of 25 km<sup>-1</sup>.

Note, that alternative, single FOV lidar methods were developed to estimate droplet microphysical properties (Donovan et al., 2015; Snider et al., 2017). However, these techniques depend on critical assumptions so that only a rough estimation of the liquid-phase microphysical properties is possible (Kalesse et al., 2016; Zhang et al., 2019). This fact was the motivation for the development of the advanced dual FOV lidar techniques (Schmidt et al., 2013, 2014; Jimenez et al., 2020a, b).

**Figure 1.** Data analysis concept to obtain the liquid-phase and ice-phase properties of a MPC layer. Dual FOV lidar observations from 50-100 m above the base of the liquid-dominated cloud layer are used to retrieve the liquid-phase properties of the MPC. Observations are averaged over 7 minutes. The results are assigned to 75 m above cloud base (CB), indicated by thick horizontal blue lines. Droplet-free conditions in the ice virga zone below CB, and thus, are used to retrieve the ice-phase properties by combining lidar and radar measurements between 100 and 400 m below CB. The 7 minute mean values of the retrieval products are assigned to 250 m below cloud base, indicated by thick horizontal red lines.

## 2.5 Lidar-radar synergy (LIRAS): microphysical properties of ice crystals





Information about ice crystal properties such as the ice crystal number concentration (ICNC) and ice water content (IWC) can be obtained from the combination of Ka-band ARM Zenith Radar (35 GHz cloud Doppler radar) and 532 nm backscatter lidar observations (Bühl et al., 2019a; Ansmann et al., 2025a). Table 1 includes the MPC ice-phase properties. For a close comparison with the liquid-phase properties 75 m above cloud base we used the ice-phase properties obtained from the lidar-radar retrievals just below the base of the liquid-dominated cloud layer (see Fig. 1). Within the virga zone, backscatter contributions by cloud droplets to total backscattering of laser photons are negligible. In contrast to ice crystals, cloud droplets do not fall out of the main MPC cloud layer and thus do not perturb the pure-ice observations and lidar-radar data analysis below the liquid-bearing MPC layer. Even in the presence of a few droplets, reaching lower heights by turbulent motions, the impact of droplets on the lidar backscatter coefficients and the radar reflectivity values in the ice-dominated virga zone and thus on the LIRAS products is negligible.

According to the airborne in situ MPC observations of Mioche et al. (2017), the ice-phase retrieval products just below the main cloud deck, represent well the ice properties in the lower half of the liquid-bearing cloud layer. In the upper half, the ice crystal properties change much with height as a function of varying ice nucleation rates and the especially strong changes of the sizes of the crystals by water vapor deposition shortly after the nucleation events.

The applied LIRAS-ice (LIdar RAdar Synergy - retrieval of ice microphysical properties) analysis scheme (Bühl et al., 2019a; Ansmann et al., 2025a) was also used in Arctic cirrus formation studies (Ansmann et al., 2025a). LIRAS-ice makes use of the measured profiles of the radar reflectivity factor Z at 8.5 mm wavelength and of the ice crystal extinction coefficient  $\alpha_{\rm ice}$  at 532 nm wavelength. Accurate 532 nm ice single-scattering extinction coefficients are of fundamental importance to retrieve trustworthy products. Very accurate  $\alpha_{\rm ice}$  profiles are obtained by means of the Raman lidar method. In the first step, the height profile of the single-scattering ice crystal backscatter coefficient is determined by using a modified version (Baars et al., 2017; Jimenez, 2021) of the Raman lidar method (Ansmann et al., 1992). In the second step, the backscatter profile is

multiplied with the well known single-scattering 532 nm ice crystal lidar ratio (extinction-to-backscatter ratio) of around 32 sr to get the required profile of the single-scattering ice crystal extinction coefficient  $\alpha_{\rm ice}$ . For more details (including references for cirrus lidar ratio observations and retrievals) we refer to Ansmann et al. (2025a) and to the articles of Bühl et al. (2019a) and Ansmann et al. (2019b).

Complementary to the dual FOV lidar products for the liquid phase the LIRAS products are the ice extinction coefficient  $\alpha_{\rm ice}$ , the effective radius  $R_{\rm e,ice}$ , IWC, and ICNC. The ice water path (IWP, vertically integrated IWC) is calculated from the IWC profile values in the ice virga plus the IWC values in the liquid-dominated cloud layer. Here, we use the virga IWC value at the top of the virga zone to be representative for the entire liquid-dominated cloud layer, i.e., we assume a height constant IWC profile from cloud base to the top of the cloud layer. The contribution of ice crystals in the liquid-containing layer to the IWP may be overestimated by about 20% when assuming an height-constant IWC profile from base to top of the liquid-bearing cloud layer (Mioche et al., 2017). Ice nucleation at cloud top and growth of the nucleated particles lead to a steady increase of IWC from cloud top (IWP=0) to the center of the liquid-dominated layer, and only in the lower half and in the main virga zone IWC is roughly height-independent according to the observations of Mioche et al. (2017).

## 2.6 Liquid-phase-related statistical analysis








In Sect. 5, we show statistical results of liquid-phase properties of Arctic liquid-bearing cloud layers, observed at heights >500 m. The analysis is solely based on the dual FOV polarization lidar observations from October 2019 to September 2020. In the first step, we removed all lidar observations that showed fog and low-level cloud signatures. These data cannot be analyzed properly because the laser light is quickly attenuated and the very strong backscatter signals partly overload the detectors. About 2630 hours (31%) of the MOSAiC measurement time period showed low clouds and fog. After applying several quality assurance procedures to the lidar observations, we selected around 360 hours of observations of well-defined stratiform cloud events, obtained during fog and low-cloud-free conditions, for our studies. The quality assurance procedure includes checks of the inter-channel constants between all four channels used to determine the two volume depolarisation ratios. Here, long data sets with clouds and cloud-free conditions were used to check the long-term stability of the counting efficiencies of the polarization sensitive channels. It was also checked that none of the lidar signal counts (in each channel) reached the saturation level of the detectors during cloud events.

In the next step, we calculated 7 minute mean signal profiles. Based on 3070 individual cloud profiles (7 minute averages) we computed the droplet-related optical and microphysical properties of the stratiform cloud layers. The data analysis covered the height range from 500 m to 7 km height. Above 7 km height, all detected cloud layers were pure ice clouds.

We distinguish pure liquid-water clouds (PL clouds) and MPCs in Sect. 5. All clouds showing virga structures below the cloud top layer were defined as MPCs. Those clouds, which produced no detectable virgae, were classified as pure liquid-water clouds. Cloud base and top heights were obtained from the lidar backscatter and depolarization observation (base heights) and from the radar reflectivity measurements (top heights). The MOSAiC radiosonde temperature data and respective interpolated temperature fields were finally used to assign cloud top temperatures, i.e., the temperatures at which heterogeneous ice nucleation usually starts. In the case of multiple cloud layers, only the lowest layer was considered in the lidar data analysis to avoid

undefined biases in the results for the upper layers introduced by attenuation of laser light and multiple scattering effects in the lowest layer.

We also show statistics regarding the temporal length of the measured cloud fields. The temporal length is given by the time an individual cloud layer needs to cross the lidar site. We defined cloud layers as single, individual layers, when they were detected at different heights. In the case of broken cloud fields (many cloud segments at the same height level), we counted a cloud field as one single cloud system if the detected cloud-free periods lasted for less than an hour. Shupe et al. (2006) counted individual cloud layers in the same way, i.e., cloud layers with gaps of < 1 h in duration were considered to be continuous. If a cloud-free period between subsequent cloud fields exceeded 60 minutes, the next cloud field, crossing *Polarstern* at that height level, was counted as a new cloud. We assume in this specific cloud length statistics that all cloud segments, separated even by 30-60 minutes, still developed at the same meteorological and aerosol conditions and, thus, should not be counted as individual, independent cloud layers.

## 3 MOSAiC free tropospheric aerosol conditions




We briefly illuminate the CCN and INP reservoirs in the lower and middle free troposphere over the Arctic during the MOSAiC year. Fig. 2 shows the aerosol particle number concentration  $n_{50,dry}$ , interpreted as proxy for the CCN reservoir, for different height levels up to the middle troposphere. The same values follow for the INP reservoir proxy,  $n_{250,d,dry}$  (per liter), in the case of a dust particle fraction of 7.5%. As discussed in Sect. 1, the reservoirs contain all activatable particles to nucleate droplets (CCN reservoir) or ice crystals (INP reservoir).

To provide an impression of the CCN- and INP-related aerosol conditions in the lower and middle troposphere, the MOSAiC time series of  $n_{50,dry}$  and  $n_{250,d,dry}$  are shown for the height levels of 1, 3, and 5 km height in Fig. 2. As mentioned, stratiform liquid-water-bearing cloud layers were observed up to 7 km height.

The high variability at all three height levels in Fig. 2 reflects the frequency of occurrence of aerosol transport events towards the central Arctic. A clear annual cycle of the particle number concentrations is visible for the 1 km height level. The values for  $n_{50,\rm dry}$  at 1 km height mostly ranged from 50-500 cm<sup>-3</sup> during the MOSAiC winter half year and from 5-150 cm<sup>-3</sup> during the short summer season from June to September 2020. During the long winter period, Arctic haze conditions dominate. As mentioned, in the summer, a mixture of anthropogenic pollution, wildfire smoke, marine particles, and mineral dust prevailed in the free troposphere over the central Arctic. In contrast to the 1 km height level, an annual cycle is absent in the time series of the particle number concentrations at 3 and 5 km height levels. The estimates of  $n_{50,\rm dry}$  and  $n_{250,\rm d,dry}$  (for a 7.5% dust fraction) at 3 and 5 km are roughly a factor of 3-10 lower than the number concentrations at 1 km height.

## 290 4 MOSAiC MPC case studies

The temporal evolution of tow long-lasting MPC events are discussed in Sect. 4.1-4.2. We selected a cold, mid-winter case (29-31 December 2019) with a text-book-like MPC evolution at Arctic haze conditions over more than 27 hours. The second

Figure 2. Particle number concentration  $n_{50,dry}$  obtained from lidar observations at cloud-free conditions for the height levels of 1 km (blue circles), 3 km (orange circles), and 5 km height (white circles). Data gaps, especially during the summer months, indicate long phases with low clouds and fog.  $n_{50,dry}$  can be regarded as a proxy for the reservoir of activatable CCN. The same values, shown in the figure for  $n_{50,dry}$ , can be obtained for  $n_{250,d,dry}$  (dust fraction of  $n_{250,dry}$ , now in number per liter) by applying the respective extinction-to- $n_{250,dry}$  conversion factor and by assuming a dust fraction of 7.5%.  $n_{250,d,dry}$  may indicate the reservoir of most favorable INPs.

case was measured in the late summer (21 September 2020). The MPC was observed over 15 hours and developed in a wildfire smoke- and anthropogenic haze-polluted environment.

## 4.1 29-31 December 2019 MPC case study




# 4.1.1 Macrophysical properties and meteorological conditions

A very stable MPC deck was observed over *Polarstern* on 29-31 December 2019. The results are presented in Figs. 3 to 5. The *Polarstern* was at 86.6°N and 116°-118°E and drifted with the pack ice. The cloud system crossed the *Polarstern* with a mainly westerly wind component. The radiosonde observations indicated horizontal wind velocities of 4-6 m s<sup>-1</sup> in the height range from the surface up to 4 km. Satellite imaginary (Worldview-earthdata-nasa-2024) showed a more than 500 km times 200 km large MPC field in the North Pole region on 30 December 2019. According to the aerosol source identification scheme of Radenz et al. (2021a), the air mass at 1-3 km height traveled mostly within the European and North American sector at latitudes >60°N during the last 10 days before reaching the MOSAiC field site. Arctic haze pollution dominated the aerosol conditions in the lower free troposphere in the central Arctic to that time (Engelmann et al., 2021; Ansmann et al., 2023).

An overview of the macrophysical cloud properties and meteorological conditions are given in Fig. 3. At midnight on 29-30 December 2019, a warm and moist air mass reached *Polarstern* at heights >1 km according to Fig. 3d and the 24 UTC radiosonde temperature profile in Fig. 3a. The radiosonde was launched at 23 UTC on 29 December 2019. RH<sub>w</sub> peaks close to

Figure 3. Temporal evolution of a long-lasting MPC event

observed with Raman lidar aboard *Polarstern* at 86.6°N, 116-118°E, on 29-31 December 2019 (panel b). The height-time display of the 1064 nm range-corrected signal in panel b shows the droplet-dominated cloud layer (in red) between 2 and 3 km height and virgae (in yellow) of falling ice crystals. Profiles of temperature (panel a) and relative humidity (over water, panel c) were measured with MOSAiC radiosondes launched at 23 UTC (29 December), 5 and 17 UTC (30 December) and 5 UTC (31 December 2019). Thin black vertical lines in panel b indicate times of radiosonde observations. Cloud top temperature was between -28 and -32°C. In panel d, the relative-humidity field derived from the water vapor Raman lidar measurements is shown. Several dark bars in panel b and white bars in panel d indicate times with non-useful lidar observations (partly caused by automated calibration and adjustment events). In panel d, the large white area above the MPCs indicates missing lidar data. In panels b and d, the lidar observations are biased in the near range (<250 m).

100% at 1.5-2 km height (see Fig. 3c) caused first cloud formation including ice nucleation and virga development at  $-24^{\circ}$ C as can be seen in Fig. 3b.

A droplet-dominated cloud layer then developed in the moist air mass between 2 and 3 km height shortly after midnight. The vertical extent of the cloud layer (in red in Fig. 3b) was 300-400 m according to the 6 UTC and 18 UTC radiosonde humidity profiles in Fig. 3c. The relative humidity over water  $RH_w$  reached 100% in the droplet-containing cloud layer. The yellow stream-like structures in Fig. 3b are produced by virgae of falling ice crystals. Almost no windshear and low windspeeds allowed an undisturbed evolution of the ice virgae. The virgae got thinner with time and the vertical extent decreased when the air mass became drier below 2 km height after 12 UTC on 30 December 2020 (see Fig. 3d). We assume that each virga is related to an individual updraft event during which CCN activation as well as ice nucleation preferably takes place (Shupe et al., 2008; Khain et al., 2022).

After the formation and establishment of an opaque liquid top layer, a strong drop in the temperature of the MPC top layer occurred, as the comparison of 23 UTC with the 5 and 17 UTC radiosonde temperature profiles in Fig. 3a indicate. The strongest temperature decrease was found at cloud top. The temperature decreased from around  $-27^{\circ}$ C (radiosonde launch at 23 UTC) to  $-32^{\circ}$ C (radiosonde launched at 5 UTC) at 2.8-2.9 km height as a result of the strong emission of long-wave radiation by the cloud top layer. The strong radiative cooling leads to decreased static stability in the cloud top layer and buoyant production of turbulence within and below the cloud top layer (Roesler et al., 2017) and to the evolution of regular pattern of updraft and downdraft periods (Shupe et al., 2008). In the continuously occurring updrafts, CCN activation and condensational growth of droplets takes place as well as ice nucleation and crystal growth by water vapor deposition. The longevity of the MPC deck is, in our opinion, the result of the continuous nucleation of new liquid water droplets.

The humidity conditions on 29-31 December 2019 over the *Polarstern* are shown in Fig. 3d. Blue and red colors indicate low relative humidity of <40% and values of 100%, respectively. The lidar measures the water-vapor mixing ratio with a vertical resolution of 7.5 m and a temporal resolution of 30 s. By using the MOSAiC radiosonde temperature profiles (available with a resolution of 6 hours), the mixing ratio profiles were converted into relative humidity profiles (see Sect. 2.2 for more details).

Pronounced lofting of the air mass occurred until about 7:00 UTC on 30 December 2019. The liquid-dominated cloud layer (in red in Fig. 3b) formed at water supersaturation conditions the formation of the cloud top layer. Strong ice nucleation and crystal growth started immediately in the droplet environment at favorable immersion freezing temperatures below  $-30^{\circ}$ C (and corresponding ice saturation ratio  $S_i > 1.34$ ) at cloud top. With decreasing temperatures during further cooling of the MPC top the ice nucleation efficiency of activatable INPs further increases (DeMott et al., 2015; Kanji et al., 2017; Wex et al., 2019). Intense and extended virgae of falling ice crystals developed and reached low heights. Sublimating ice crystals kept the relative humidity high between the cloud top layer and 1 km height until 14:00 UTC. Cooling of the air during the sublimation processes may contribute to a strong downward motion of ice-containing air layers. With time the advected air mass became significantly drier. However, sublimation of crystals were able to keep the relative humidity high in a 300-400 m thick layer below the droplet-dominated top layer. The released water vapor, CCNs, and INPs may have partly been transported back into the cloud top layer during updraft periods. Further CCNs and INPs may have entered the cloud from above, from below via blowing snow effects (Gong et al., 2023), or were advected within the moist air mass with the prevailing westerly windflow.

The cloud top layer became thinner (optically and geometrically) so that water vapor observations were possible again (as in the beginning) up to 4-5 km height. The MPC dissolved at 4:00 UTC. The 6 UTC radiosonde (launched at 5 UTC) indicated water subsaturation at all heights and maximum values around 90% at 2.5 km height.

## 4.1.2 Microphysical properties: ice phase vs liquid phase








The retrieval products for the liquid and the ice phase are presented in Fig. 4. The three height-time displays in Fig. 4a, 4b, and 4c show the basic lidar and radar observations. The cloud radar provides an overview of the temporal evolution of the MPC system in terms of the radar reflectivity Z (caused by ice crystals at 8.5 mm wavelength in Fig. 4a). The radar detects all ice crystals in the vertical tropospheric column. Even the ice virgae originating from upper tropospheric cirrus clouds above the MPC are detected. These crystals may have influenced the MPC evolution via the feeder-seeder effect (Rauber and Tokay, 1991; Ansmann et al., 2008, 2009; Ramelli et al., 2021) before 7 UTC on 30 December 2019. In seeder-feeder processes, ice crystal seeds enter a liquid cloud layer from above and grow fast at the expense of liquid water droplets which evaporate during the ice growth process.

According to the radar observations, the top height of the MPC was at about 3 km height. The ice crystals, which were nucleated at the top of the liquid-bearing cloud layer, grew and fell out of the MPC layer and formed large virgae. Since the radar is most sensitive to the ice crystals, which are distributed over the entire MPC height range from cloud top to the virga base height, the base of the liquid-dominated cloud layer remains undetected in the radar reflectivity height-time display in Fig. 4a. The black dots, shown in Fig. 4a are indicating the base of the liquid-containing layer of the MPC. This information is taken from the lidar observations.

The lidar precisely detects the base of the liquid-dominated cloud layer, given also as black dots in Fig. 4b. However, the lidar is usually unable to detect the top of this cloud layer because of strong attenuation of laser light by droplet scattering processes. As mentioned in Sect. 2.4, cloud top heights can be detected as long as the cloud optical depth  $COD \le 2.5$ . However, in most cases COD is > 5. The ice extinction coefficient in the ice virgae (at 532 nm wavelength, below the black dots in Fig. 4b) is obtained by multiplying the highly resolved backscatter coefficients with a typical ice crystal lidar ratio of 32 sr (see discussion in Ansmann et al. (2025a, b)).

Figure 4c shows the observations of the lidar volume depolarization ratio. The depolarization ratio permits a clear separation of the droplet-dominated cloud layer and the ice crystal virga zone. The volume depolarization ratio is usually close to zero at cloud base and monotonically increases towards 0.2 with increasing penetration of the laser pulse into the liquid-dominated cloud layer. In contrast, the volume depolarization ratio is 0.4-0.6 for ice crystal backscattering. In the lowest parts of the virgae, where ice crystals sublimate, the impact of clear-air depolarization, causing depolarization ratios around 0.01, steadily increases so that the volume depolarization ratio decreases towards the respective clear-air value.

When zooming into the virga zone in Fig. 4a to 4c, singular, individual virgae show pronounced backscattering and depolarization features and thus can easily be identified. The regularly occurring virga structures are the result of enhanced ice production during the updraft periods (Shupe et al., 2008). During upwind phases, the conditions for both CCN activation and droplet growth as well as ice nucleation and crystal growth are most favorable, while during downdraft periods, only ice crystal

growth (Wegener-Bergeron-Findeisen (WBF) mechanism) (Korolev, 2007) and sedimentation continues to occur (Khain et al., 2022). We counted 30-35 individual virga structures within 6 hour intervals (12-18 UTC, 18-24 UTC on 30 December 2019), i.e, updrafts occurred every 10-12 minutes. These structures are also visible in the respective height time display of the cloud radar mean Doppler velocity (not shown). Cycles of up- and downdrafts, superimposed on particle fall velocities, are well resolved in the Doppler radar observations. An example is shown in Sect. 4.2.








Taking the radiosonde observations of the horizontal wind speed of about 5 m s<sup>-1</sup> up to 4 km height into considerations, the updraft phases were horizontally separated by about 3 to 3.5 km from each other. Similar horizontal scales of the updraft-downdraft periods were observed by Shupe et al. (2008). Note that during the late summer MPC event, observed on 21 September and discussed in Sect. 4.2, 40-60 virgae occurred within six hours and point to horizontal separations of updraft zones by only 1.8-2.2 km. Horizontal wind velocities were again around 5 m s<sup>-1</sup>. The difference between the winter and late summer observations may be related to different meteorological and cloud optical properties which determine the strength of cloud top cooling and thus the up and downdraft characteristics. After formation of the MPC deck the cloud top temperature decreased by 5 K on 30 December 2019 and by 2 K on 21 September 2020.

In Fig. 4d to 4g, the retrieval products for both the liquid and the ice phase are given. The liquid-phase values were obtained from the dual-FOV-lidar observations around 75 m above cloud base (see sketch in Fig. 1) and the ice extinction coefficient and the microphysical properties for the ice phase were retrieved from the combined lidar-radar observations around 250 m below the base of the droplet-dominated cloud layer. The results in Fig. 4 show that the main phases of the entire life cycle of the MPC system were well resolved. We are aware of the fact that caution has to be exercised in studies of cloud life cycles (Lagrangian perspective) by analyzing Eulerian measurements (measurements at a fixed location) (Fridlind and Ackerman, 2018; Grabowski, 2020; Khain et al., 2022)). Three phases of the MPC evolution can be distinguished. During the initial phase with large-scale lofting until 8:00 UTC on 30 December 2019, the optical and microphysical properties (except the effective radius) of both phases are quite variable. This variability is probably partly related to the potential impact of seeder-feeder effects. The radar observations in Fig. 4a show a strong virga field above the MPC until 7 UTC so that ice crystals could in principle enter the MPC from above (as ice seeds). This additional ice production may have caused the intensification of ice virga backscattering during the initial phase of the MPC lifetime. However, the top of the liquid-bearing cloud layer increased until 7:00 UTC as well. The increase of the top height is associated with a further decrease of the cloud top temperature and corresponding increase of the ice nucleation efficiency of INPs, resulting in increasingly strong ice crystal nucleation and overall ice production and thus intensification of virga structures. The ice crystal extinction coefficient increased as long as the cloud top height increased. Afterwards, the ice extinction coefficient decreased quickly until 8-9 UTC and then remained almost constant until 23 UTC.

The main and second phase of the life cycle (from about 8:00 UTC to 23:00 UTC) was characterized by a stable liquid-dominated cloud deck which controls the further evolution of the MPC system. The third and final phase of the MPC life cycle, occurred after 23:00 UTC on 30-31 December 2019. The water vapor content of the advected air mass decreased and caused the dissolution of the cloud deck, as already mentioned above and shown in Fig. 3.

Figure 4. Temporal evolution of the winter MPC as observed (a) with cloud radar (radar reflectivity Z) and (b, c) with lidar (panel b: particle extinction coefficient  $\alpha_{ice}$ , panel c: volume depolarization ratio  $\delta_v$ ) aboard *Polarstern* at 86.6°N on 30-31 December 2019. Black dots in (a) and (b) show the base of the liquid-bearing cloud layer as detected with lidar. Time series of (d) the droplet extinction coefficient (in blue) and the ice crystal extinction coefficient (in red), (e) the effective radii of droplets (blue) and ice crystals (red), (f) LWC (blue) and and IWC (red), and (g) CDNC (blue) and ICNC (red) are retrieved by using the basic observations in panels (a), (b), and (c). The droplet properties (in blue) are computed for the height of 75 m above the base of the liquid-dominated cloud top layer. The ice properties (in red) are determined in the virga zone around 250 m below the base of the liquid-dominated cloud layer. SD bars indicate the uncertainty in the retrieval products. Gaps in the time series, associated with white vertical columns in (a) to (c) indicate missing data.

During the stable period, the liquid-phase and ice-phase extinction coefficients show values around  $10 \text{ km}^{-1}$  and 0.1- $0.2 \text{ km}^{-1}$ , respectively. The effective droplet radius slowly decreased from  $8 \mu \text{m}$  to  $5 \mu \text{m}$ , and the ice crystal effective radius was around  $40 \mu \text{m}$ . LWC and IWC values were on average  $0.05 \text{ g m}^{-3}$  and  $0.003 \text{ g m}^{-3}$ , respectively. For the number con-

centrations, CDNC and ICNC, we obtained values from 30-150 cm<sup>-3</sup> and between 0.5 and 1 L<sup>-1</sup>, respectively. In terms of LWC and IWC, the ice-phase fraction IWC/(IWC+LWC) (Korolev et al., 2017) was thus about 0.06. In terms of the optical properties  $\alpha_{\text{liq}}$  and  $\alpha_{\text{ice}}$ , the ice-phase fraction  $\alpha_{\text{ice}}/(\alpha_{\text{liq}} + \alpha_{\text{ice}})$  was 0.01-0.02. When using the number concentrations, CDNC and ICNC, we end up with an ice phase fraction on the order of  $10^{-5}$  to  $10^{-6}$ . That means only a very small fraction of the droplets was involved in immersion freezing processes. INPs may have been entrained continuously from above and from below or advected with the main air flow. A depletion of the INP reservoir is not visible.






By using the observations of  $n_{50,\rm dry}$  at 3 km height in Fig. 2 for the period from 25 December 2019 to 12 January 2020 (around day of the year = 0), the CCN reservoir indicates particle number concentrations of 20-50 cm<sup>-3</sup>. The CDNC values of 30-150 cm<sup>-3</sup> suggest that even smaller particles with radius <50 nm were activated to form liquid droplets. When assuming that 1% of CCN reservoir consisted of dust particles and that these dust particles defined the INP reservoir for immersion freezing this INP reservoir contained activatable INPs of the order of 200-500 L<sup>-1</sup>. The most favorable INPs are the larger dust particles. According to Fig. 2 we estimated values of 20-50 L<sup>-1</sup> for  $n_{250,d,dry}$  assuming a dust fraction of 7.5%. For a dust fraction of 1%, we obtain an INP reservoir of most favorable INPs of about 2.5-6.5 L<sup>-1</sup>. The retrieved ICNC values were most of the time 0.5-1 L<sup>-1</sup>. We hypothesize that the CCN and the INP reservoirs were continuously refilled by vertical transport processes from below and above the MPC layer and thus were always well filled.

The most surprising observation is the steady increase of the CDNC from values around 30 cm<sup>-3</sup> at 11:00-11:30 UTC to values above  $100 \text{ cm}^{-3}$  at 19:00 UTC, and, at the same time, the decrease of the effective droplet radius from 8 to 5  $\mu$ m. This is surprising because the continuously occurring processes of ice growth by water vapor deposition, associated with shrinking and evaporation of droplets, should lead to a reduction of the droplet number concentration and, in addition, to a decrease of the effective droplet radius. Droplet-droplet collision and coalescence processes may cause a less strong reduction of the effective radius, but contribute to a reduction of CDNC. The potential impact of riming processes is not discussed here. These processes seem to be unimportant at temperatures  $< -20^{\circ}$ C (Waitz et al., 2022).

It cannot be excluded that the observed changes in the CDNC and  $R_{\rm e,liq}$  numbers may be simply related to changing aerosol conditions, however, to our opinion, the observation of an increasing CDNC and, simultaneously, decreasing effective droplet radius  $R_{\rm e,liq}$  with time is a clear sign that CCN activation occurred and persistently refilled the small droplet fraction and stabilized in this way the broad droplet size distribution. Our hypothesis of a strong role of droplet nucleation in the lifetime of MPCs is in line with model results of Khain et al. (2022). Based on simulations with their 2D Lagrangian-Eulerian mixed-phase cloud model, Khain et al. (2022) found in addition that ice particles at concentrations < 1 L<sup>-1</sup> do not affect the LWC in the liquid-dominated cloud top layer significantly. Cloud glaciation (via the WBF mechanism) is possible when ICNC exceeds  $10 L^{-1}$ , i.e., only when ICNC >10 L<sup>-1</sup>, the WBF process has a strong impact on the conversion of cloud liquid water to ice water. As can be seen in Fig. 4g, ICNC was mostly in the range of 0.3-1 L<sup>-1</sup>. A comparably weak WBF process and a correspondingly weak reduction of the relative humidity caused by water vapor deposition on ice crystals contributes to favorable conditions for CCN activation.

An interesting observation was made between 22 and 23 UTC in lowest troposphere at heights below 700 m. The radar and the lidar detected reflection and depolarization features that pointed to the presence of ice crystals. The  $RH_w$  profile of the

radiosonde launched at 23 UTC showed high humidity values close to 100% and temperatures of -24°C. Formation of rather thin water clouds and subsequent immersion freezing and growth of ice crystals as well as strong growth of ice crystals falling into this ice-supersaturated layer from above may have been responsible for the observed ice crystal backscattering features.

As mentioned in the introduction, the parameterization of ice nucleation, i.e., here immersion freezing, is presently an important aspect in the discussion how to consider ice nucleation in atmospheric models. Long-lasting MPC events as observed on 29-31 December 2019 are favorable cases to demonstrate (by using a large-eddy-simulation-informed MPC aerosol-cloud model) that a time-dependent ice nucleation mechanism paired with a large INP reservoir can sustain continuous ice crystal production for several hours (Knopf et al., 2023). The longevity of such clouds, sustaining both liquid and ice crystal formation over a long time period, is poorly represented across global climate models when using a time-independent approach to characterize INP activation. As outlined in Sect. 1, the underlying freezing parameterization defines the number of ice-nucleating particles (INPs) available for ice formation, termed INP reservoir. A time-dependent freezing description yields a substantially greater INP reservoir than time-independent approaches, and therefore greater ice formation over long time periods. The observed low ICNC values, pointing to low ice nucleation rates, and the obviously sufficiently large INP reservoir lead to the recommendation to use a time-dependent immersion freezing parameterization in MPC simulations.






Figure 5 finally presents height-time displays of ICNC (panel a) and IWC (panel b) together with time series of the liquid water path LWP and the ice water path IWP in Fig. 5c. The retrieval of LWP and IWP was described in Sect. 2.1 and 2.5, respectively. ICNC and IWC decreased with decreasing height in the virga zone during the stable phase of the life cycle after 12:00 UTC. This height dependence is the result of sublimation of crystals. As long as the virgae do not reach the ground the released water vapor and particles are available to be transported back into the cloud-top layer and to act again as CCNs and INPs. The IWP ranges from 10-35 g m<sup>-2</sup> during the initial phase of the MPC observation (before 11 UTC) and was very constant for the rest of the MPC lifetime with values around 5 g m<sup>-2</sup>. The LWP was most of the time <35 g m<sup>-2</sup>, and around 15 g m<sup>-2</sup> during the second and third phase of the MPC lifetime. In terms of LWP and IWC, the ice-phase fraction (IWP/(IWP+LWP)) was thus on the order of 0.25, or 0.2 when considering a potential overestimation of IWP by 20% into account as discussed in Sect. 2.5.

The observed properties for this mid-winter MPC event agree well with respective findings obtained from airborne in situ measurements conducted in the framework of several Arctic aircraft campaigns in the spring seasons of 2004, 2007, 2008 and 2010 north and around Svalbard (78°N, 15°E), Longyearbyen, Norway and Kiruna (68°N, 20°E), Sweden (Mioche et al., 2017). The MOSAiC December MPC showed slightly lower values compared to airborne in situ spring observations. In numbers, the spring time Arctic MPCs showed droplet extinction coefficients of  $20\pm10~\mathrm{km^{-1}}$ , a droplet effective radius around 7.5  $\mu$ m, and LWC and CDNC values of  $0.2\pm0.1~\mathrm{g}~\mathrm{m}^{-3}$  and  $120\pm60~\mathrm{cm^{-3}}$ , respectively. For the ice phase, Mioche et al. (2017) found extinction coefficients of about  $0.4\pm0.3~\mathrm{km^{-1}}$ , effective ice crystal radius of  $50\pm25~\mu$ m, IWC of  $0.03\pm0.03~\mathrm{g}~\mathrm{m}^{-3}$  and ICNC of  $3\pm2~\mathrm{L^{-1}}$ .

The observations on 30-31 December 2019 agree also well with long-term lidar-radar observations over Leipzig from 2011 to 2015. Bühl et al. (2016) shows LWP, IWC, and and ice phase ratio (IWC/LWC) as a function of the MPC top temperature. At -30°C, IWC ranged from  $10^{-6}$  to  $5 \times 10^{-6}$  kg m<sup>-3</sup>, LWP was on the order of 0.01 kg m<sup>-2</sup> for MPCs with cloud top

**Figure 5.** Evolution of the MPC observed on 30-31 December 2019 in terms of (a) the ice crystal number concentration (ICNC), (b) ice water content (IWC), (c) liquid water path LWP (blue) and ice water path IWP (red). Error bars show the uncertainty in the retrieved LWP and IWP values. White vertical columns indicate periods without lidar data. The uppermost red rim of the data field in panel a is a bias and occurs at the base (sharp edge) of the liquid-dominated cloud layer. The black crosses indicate the height of 250 m below cloud base for which the time series of the ice-phase microphysical properties in Fig. 4 are shown.

temperatures of  $-30^{\circ}$ C, and the respective ice phase ratio IWC/LWC showed values around 0.03-0.05. During the stable phase of the MPC life cycle we observed LWP of 0.015 kg m<sup>-2</sup> on 30 December 2019, IWC of  $3 \times 10^{-6}$  kg m<sup>-3</sup>, and for the IWC/LWC ratio we obtained 0.003 to 0.06.

## 4.2 21 September 2020 MPC case study



In this section, we discuss the temporal evolution of a long-lasting late-summer MPC event that was observed near the North Pole on 21 September 2020. *Polarstern* was drifting within an ice field at 88.5°N. The results for this case are given in Figs. 6-8. Similar to the air mass transport on 29-31 December 2020, the free tropospheric air mass, in which the MPC developed, crossed areas mainly north of 60°N (Northern Canada, Iceland region) during the last 10 days before arriving over *Polarstern*. In contrast to the December 2019 case study, the cloud top temperatures were around  $-22.5^{\circ}$  to  $-23.5^{\circ}$ C (at 3.3-3.4 km height) according to the radiosondes launched at 11 and 17 UTC on 21 September 2020. The cloud layer was thus warmer by 8-10 K than the one in December 2019. A drop by 2 K at 3.3 km height from 5 UTC (cloudless conditions) to 11 UTC was

**Figure 6.** Perturbations of the MPC development on 21 September 2020 at 12:00 UTC, visible in the lidar observations of the volume depolarization ratio in (a) at heights of 3-3.5 km. In (b), the cloud radar reflectivity Z and, in (c), the mean Doppler velocity derived from the Doppler radar observations are shown. The gray to orange colors in panel c may indicate upwind areas when taking an average ice crystal sedimentation speed of 50 cm s<sup>-1</sup> into account.

observed due to radiative cooling of the liquid-dominated cloud top region. Anthropogenic pollution and wildfire smoke from Siberia and North America were advected into the central Arctic since mid-July 2020 (Boyer et al., 2023; Heutte et al., 2024; Ansmann et al., 2023). It is reasonable to assume that, in addition, soil dust and mineral dust, probably predominantly from nordic sources (e.g., from Iceland) as well as biological aerosol particles (locally produced or advected from the surrounding continents) contributed to the Arctic aerosol mixture. The observed CDNC values of >100 cm<sup>-3</sup> reflect the polluted conditions.



A strong perturbation of the cloud evolution occurred at 12:00 UTC. This noon event is highlighted in Fig. 6. The reason for the perturbation, which influenced the further development of the MPC system for hours, is unknown and will not be discussed here. The base height of the liquid-bearing cloud layer was lofted by 150-200 m within a few minutes at 12:00 UTC as can be seen in Fig. 6a. The 12:00 UTC perturbation obviously triggered an intensification of ice production and broadened the virga structures. A significantly stronger backscattering by ice crystals in the virga zone was detected after 12:10 UTC in the lidar data, and after 12:20 UTC in the radar data. The virga activity became weaker again after 14 UTC.

The strong perturbation at 12:00 UTC is not directly visible in the cloud reflectivity and Doppler velocity height-time displays in Fig. 6b and c. The radar reflectivity Z increased in the virga zone belwo 3 km height after 12:20 UTC. The delay can be explained by ice crystal sedimentation with crystals falling speeds of 20-60 cm s<sup>-1</sup>. If ice nucleation occurs at the top

**Figure 7.** Same as Fig. 4, except for a late summer MPC event, observed on 21 September 2020. *Polarstern* was at 88.5°N. Black dots in (a) and (b) show the base of the liquid-dominated cloud layer. A strong perturbation of the MPC evolution occurred at 12:00 UTC.

of the approximately 300 m thick liquid-bearing cloud layer, it takes about 10-20 minutes before the nucleated and steadily growing ice crystals reach the cloud base and influence the virga structures below the main cloud deck as seen by the radar.


In Fig. 6c, many up- and downdraft phases are visible in the height-time display of the mean Doppler velocity. The high frequency of occurrence of virga structures in Fig. 6b between 12:20 and 14:10 UTC is well reflected in the frequency-of-occurrence of periods with Doppler velocities <-0.5 m s<sup>-1</sup> in Fig. 6c. Strong downdraft motion together with crystal sedimentation often caused Doppler velocities <-1 m s<sup>-1</sup>. The color scale is adjusted such that Doppler velocities >-0.5 m s<sup>-1</sup> may already indicate updraft periods assuming mean ice crystal fall speed of about 0.5 m s<sup>-1</sup>.

Figure 7 presents an overview of the entire 21 September MPC observation in terms of the liquid and ice-phase products. The 12 UTC perturbation triggered strong ice nucleation in the liquid-dominated cloud layer. Cooling of the ascending air parcels obviously significantly increased the ice nucleating efficiency of the INPs. The freshly formed ice crystals grew fast and fell out of the liquid-containing cloud deck and caused the observed strong virga activity after 12 UTC. The ICNC in the virga zone at 250 m below the base of the liquid-bearing cloud layer increased from values of 0.5-1 L<sup>-1</sup> before 12 UTC to 2-3 L<sup>-1</sup> shortly after 12 UTC. The respective ice extinction coefficient increased from 200 Mm<sup>-1</sup> to 500-600 Mm<sup>-1</sup>. Accordingly, also the IWC increased after the perturbation.







The CDNC dropped strongly from  $180\text{-}200~\text{cm}^{-3}$  before the perturbation at 12~UTC to  $60\text{-}80~\text{cm}^{-3}$  after 12~UTC within a few minutes. The CDNC further decreased to values of  $30~\text{cm}^{-3}$  at 13~UTC and was thus almost an order of magnitude lower than the number concentration before 12~UTC. The most reasonable explanation for this strong reduction of the droplet number concentration is ice formation at the expense of the liquid water droplets via the the WBF mechanism. It is less likely that enhanced turbulence caused by the perturbation initiated strong droplet collision and coagulation processes and thus a respectively strong reduction of the droplet population. However, besides ice production also an increase of the LWC was observed. During the lofting event most probably condensational growth of the remaining droplets occurred. This hypothesis is in line with the increase of the effective droplet radius from values close to  $5~\mu$ m before 12 UTC to  $7~\mu$ m after 12 UTC and even to values  $>10~\mu$ m later on. The strong variability of the extinction coefficients, CDNC, ICNC, IWC, and LWC points to quite unstable conditions for more than two hours after the perturbation at 12~UTC.

It is noteworthy to mention that the effective radius of the ice crystals was quite constant from 10 UTC to 21 UTC with values around 50  $\mu$ m. We may conclude that the crystals grew by about 0.1  $\mu$ m s<sup>-1</sup> (in diameter) during the roughly 1000 s long sedimentation period from cloud top to 250 m below the base of the liquid-bearing cloud layer, assuming a crystal sedimentation speed around 0.5 m s<sup>-1</sup> and the absence of significant collision and aggregation processes. This growth rate is in good agreement with the laboratory studies of Bailey and Hallett (2012).

An impressive observation is the recovery of the CDNC during the afternoon hours. The CDNC dropped from maximum values of 200-300 cm<sup>-3</sup> at 11:00 UTC to values around 30 cm<sup>-3</sup> at about 13:00 UTC, and then recovered within 2.5 hours, indicated by values of 200-400 cm<sup>-3</sup> after 15:30 UTC. For the effective droplet radius, we found values of 5  $\mu$ m at 11:00 UTC, maximum values close to 15  $\mu$ m at about 13:00 UTC, and then later on again values around or just below 5  $\mu$ m at 16:00 UTC. These strong and coherent changes in the CDNC and  $R_{\rm e,liq}$  from 13:00 to 16:00 UTC can only consistently be explained when taking strong CCN activation into consideration.

After 16:00 UTC, i.e., after the complete disappearance of all perturbation effects and the establishment of a new, stable equilibrium, the ice extinction coefficients showed values around 0.1-0.15 km<sup>-1</sup> and the ice crystal effective radius was around 50  $\mu$ m. LWC and IWC values were about 0.08-0.1 g m<sup>-3</sup> and 0.003-0.005 g m<sup>-3</sup>, respectively. For the number concentrations, CDNC and ICNC, the values ranged mostly from 200-300 cm<sup>-3</sup> and between 0.1 and 1 L<sup>-1</sup>, respectively. In terms of LWC and IWC, the ice-phase fraction was thus about 0.04-0.05. By using the optical properties  $\alpha_{liq}$  and  $\alpha_{ice}$ , the ice-phase fraction was 0.006. With the number concentrations, CDNC and ICNC, we end up with an ice phase fraction on the order of  $2 \times 10^{-6}$ .

Compared to the December 2019 MPC case with CDNC values of 30-150 cm<sup>-3</sup>, the CDNC values of 200-400 cm<sup>-3</sup> on 21 September indicated roughly a factor of 3-5 larger reservoirs of CCN and INPs. The dissipation of the cloud deck close to 24 UTC was caused by a steady decrease of the relative humidity at heights around 3 km as indicated by the MOSAiC radiosondes.

Figure 8 shows time series of LWP and IWP (in panel c), together with height-time displays of ICNC (panel a) and IWC (panel b). The LWP time series is widely a function of the MPC evolution above 3 km height. However, the monotonic increase of LWP from 60 to 120 g m<sup>-2</sup> between 11-12 UTC was caused by another thin liquid cloud layer that developed between 850 to 1150 m according to the 12 UTC radiosonde observation, launched at 11 UTC. This lower cloud layer obviously disappeared when the 12 UTC perturbation occurred and ice crystals at lower height were detected (see Fig. 8a and b) after 12 UTC. Seeder-feeder effects at low heights and rapid liquid-to-ice conversion may have lead to the removal of the lower liquid-containing cloud layer and therefore to the abrupt change of LWP from 120 g m<sup>-2</sup> back to values around 60 g m<sup>-2</sup> within a few minutes after 12 UTC.

After the phase with strong virga activity, i.e., after 14 UTC, the LWP increased again. The reason may be the reduced conversion of liquid water to ice. The last strong ice formation event, indicated by a pronounced virga, occurred around 15 UTC. After 16 UTC, the LWP started to monotonically decrease. The air mass became drier and the MPC deck began to dissolve slowly but steadily.

LWP decreased from values as high as 90 g m $^{-2}$  at 16:00 UTC to values around 30 g m $^{-2}$  at 18:00 UTC. IWC, on the other hand, was already mostly < 10 g m $^{-2}$  after the perturbation period ending at 14:00 UTC and remained low for the rest of the MPC lifetime. Accordingly, the ice-phase fraction was about 0.05 (at 16:00 UTC) and 0.15 (at 18:00 UTC) in terms of LWP and IWP. The precipitation signatures at heights below 1 km on 21 September show low ICNC values close to 0.01 L $^{-1}$  and IWCs close to 0.0001 g m $^{-3}$ . This is the reason why these features remained undetected in the lidar observations shown in Fig. 6a.

## 5 MOSAiC statistics of liquid-phase properties of Arctic clouds




Silber and Shupe (2022) analyzed 1362 RH profiles measured with MOSAiC radiosondes between 4 October 2019 and 19 September 2020. The authors found that 997 RH profiles (73%) contained signatures of liquid-bearing cloud layers and in 50% of these profiles (37% out of all radiosonde profiles) multi-layer cloud structures were found, i.e., liquid-bearing cloud layers from the surface up to the middle free troposphere.

Our continuous lidar observations aboard *Polarstern* provide complementary information. In about 2630 h (31%) out of the 8450 MOSAiC measurement hours from 4 October 2019 to 21 September 2020, near-surface low-level clouds and fog were observed. In the winter half year (until 1 Aoril 2020), 17% out of all observations contained low-level cloud and fog features. During the summer half year, the fraction increased to 46%. Such a high low-cloud and fog occurrence frequency (at heights <500 m) is typical for the Arctic in summer (Achtert et al., 2020; Griesche et al., 2024a). According to Achtert et al. (2020), the cloud occurrence frequency drops from values close to 50% to 20-25% for liquid-bearing clouds at heights >500 m.

**Figure 8.** Same as Fig 5, except for 21 September 2020. After a continuous increase of the LWP from 10:00 to 12:00 UTC, a strong drop of the LWP occurred at 12:00 UTC (panel c), when the strong noontime cloud perturbation occurred.



The cloud statistics presented here is based on lidar observations during the 5820 hours without fog and low clouds. We analyzed 360 h of stratiform pure liquid and mixed-phase cloud observations. The observations during these 360 hours fulfilled all signal quality criteria stated in Sect. 2.6. A measurement example with complex cloud formation between 500 m and 6.5 km height is shown in Fig. 9. The *Polarstern* was at 82°N. Layered pure liquid (PL) clouds were detected at heights 

Figure 9. Lidar observations of liquid water clouds (before 23:30 UTC of 17 June 2020) and MPCs with strong virga evolution on 18 June until 11:00 UTC. Range-corrected 532 nm backscatter signals and volume depolarization ratio are shown in panels (a) and (b), respectively. White dots in panel a indicate the cloud base height of cloud segments for which the cloud droplet number concentration (CDNC, panel c) and the effective droplet radius  $R_{\rm e,liq}$  (panel d) is derived from the dual-FOV polarization lidar observations. Error bars indicate the uncertainty in these products.

analyzed 147 cloud events, i.e., of the time periods the individual clouds fields needed to cross *Polarstern*. The 147 cloud systems covered about 500 h, of which 360 h showed cloud signatures and 140 h covered the cloud-free gaps of 

**Figure 10.** Frequency distribution of individual, coherently observed cloud layers as a function of the time period needed by the cloud field to cross the *Polarstern*. We analyzed 147 individual cloud fields, measured during the MOSAiC year from October 2019 to September 2020.

the statistical distributions of cloud base heights and estimated cloud top temperatures are shown in addition. We distinguish between cloud events with ice virgae (MP, red histograms) and without ice virgae (PL, blue histograms). The black histogram lines consider all 3070 cloud profiles. The PL cloud fraction includes all cloud layers with top temperatures > 0°C, at which ice nucleation is impossible, and clouds with top temperature < 0°C but no virga evolution. The microphysical properties of the PL clouds as LWC and the droplet extinction coefficient are strong functions of cloud temperature, which determines the amount of water vapor convertable into cloud liquid water. The cloud extinction coefficient also depends on the availability of CCNs. The red histograms show the characteristics of the liquid phase of mixed-phase clouds when all relevant ice-formation-related processes (ice nucleation, ice crystal growth, and droplet evaporation, etc.), as described in the foregoing sections, influence the cloud properties in addition.

The histograms of the PL cloud properties in Fig. 11a, b,d, and e are all at least slightly broader, and some considerably broader than the respective frequency-of-occurrence distributions for MPCs. This is related to the broader temperature distribution for PL clouds (see Fig. 11f) compared to the respective temperature histogram for MPCs and the shift of the PL-related temperature histogram towards higher temperatures. Note, that the temperature at cloud base, where CCN activation starts, is roughly 2-3 K higher than the temperature at cloud top (in panel f), where ice nucleation typically begins.


In Fig. 11a and b, the basic liquid-phase products, retrievable from the dual FOV lidar observations, are shown. The black curve, considering all clouds, shows a broad maximum of the extinction coefficient histogram with values from about 5 km<sup>-1</sup> to about 45 km<sup>-1</sup>. The PL extinction distribution is much broader than the respective MPC extinction distribution. In the case

**Figure 11.** MOSAiC pure liquid (PL, blue) and mixed-phase (MP, red) clouds statistics. The statistics cover the liquid phase of clouds with base height >500 m in terms of (a) the droplet extinction coefficient, (b) effective droplet radius, (c) base height of the droplet-dominated cloud layer, (d) LWC, (e) CDNC, and (f) cloud top temperature. The normalization (to 1.0) is applied to all clouds (PL+MP, in black). Median, mean, and SD values are given as numbers. The thick black histogram lines are based on 3070 cloud data sets, the blue histograms (PL clouds) on 1721 data sets, and the red histograms (MP clouds) on 1349 data sets.

of ice-containing clouds, the number of clouds with high droplet extinction coefficients  $> 35 \text{ km}^{-1}$  is significantly lower when compared to the respective numbers in the PL extinction histogram. The droplet extinction coefficients accumulated in the relatively narrow range from about 10-30 km<sup>-1</sup>, probably the result of the WBF process, causing the evaporation of droplets, in combination with CCN activation and new droplet formation.



The PL- and MPC-related histograms are not very different in the case of the droplet effective radius. According to the black (PL+MP) histogram in Fig. 11b, a pronounced maximum from 3-7  $\mu$ m is found. Most PL clouds and MPCs have LWCs between 0.03 and 0.13 g m<sup>-3</sup> (Fig. 11d). However, there is a pronounced right wing in the distribution in the case of PL clouds, and even a shift of the MPC-related LWC distribution, compared to the PL-related distribution, towards smaller values. This distribution shift and the broad shape of the histogram reflect the impact of the cloud temperatures (see the distributions in Fig. 11f). A broad cloud temperature distribution is obtained with values from  $-22^{\circ}$ C to  $+12^{\circ}$ C in the case of PL cloud tops, whereas most cloud top temperatures are found between  $-16^{\circ}$ C and  $-32^{\circ}$ C in the case of MPCs. In the overlap region, the total amount (PL+MP) clouds shows a clear maximum around  $-20^{\circ}$ C.

The CDNC histograms in Fig. 11e reflect the broad spectrum of CCN concentrations, i.e., aerosol conditions from very clean to heavy polluted Arctic haze and wildfire smoke situations. The CDNC histogram for pure liquid clouds spans over more than two orders of magnitude, from  $<10 \text{ cm}^{-3}$  to about  $1000 \text{ cm}^{-3}$ . Again the CDNC distribution is more narrow when ice crystal nucleation and the WBF process (ice crystal growth) come into play. The overall (PL+MP) CDNC distribution shows a pronounced maximum from 10 to  $500 \text{ cm}^{-3}$ .







Zhang et al. (2019) presented year-round lidar observations in Alaska (Utqiagvik, October 2013 to January 2017) and in Antarctica (Mc Murdo, December 2015 to January 2017). Their lidar-derived CDNC values were highest in winter (values accumulated in the range from  $30-150 \, \mathrm{cm}^{-3}$ ) and lowest in the summer with most values between 15 to  $40 \, \mathrm{cm}^{-3}$  for the Arctic station. Zhang et al. (2019) also presented lidar estimates for the effective radius  $R_{\mathrm{e,liq}}$  in Arctic and Antarctic cloud layers. Most Arctic values were between 10 and  $16 \, \mu \mathrm{m}$  in summer and 4 and 9  $\mu \mathrm{m}$  in winter.

Figure 11c provides information about the heights at which the analyzed shallow cloud layers were typically observed. Most PL cloud layers are found below about 1800 to 2000 m height (with base heights below 1500 m). PL clouds with base heights above 1500 m up to 4200 m are less frequently observed. In contrast, most free tropospheric MPCs were detected at heights above 1000 m up to 3200 m. The highest clouds showing a liquid phase were observed above *Polarstern* at about 6.5 km height (as shown in Fig. 9). Silber and Shupe (2022) provide results of backward trajectory cluster analysis and air mass origin studies for all MOSAiC liquid-bearing cloud layers. Most of the air masses originated from the North Atlantic, Europe, and Asia, mostly from latitudes > 50°N.

According to Griesche et al. (2021), our cloud statistics cover the so-called surface-decoupled fraction of Arctic stratiform clouds. These clouds develop in lofted aged aerosol layers reaching the Arctic after long-range transport from the surrounding continents and are not or only weakly influenced by local surface emissions. The main INP type in these lofted layers of complex aerosol mixtures is mineral dust (Carlsen and David, 2022). Biological particles and particles carrying biogenic or biological material also contribute to ice nucleation in the free troposphere during the short Arctic summer season from May to August (Carlsen and David, 2022). In the lowermost 500-600 m of the atmosphere (e.g., in the surface-coupled height regime) local particles with biogenic substances may dominate ice nucleation processes in the low-level clouds during the few summer months (Griesche et al., 2021; Creamean et al., 2022). The mineral dust fraction of Arctic haze controls heterogeneous ice nucleation even in the lowest part of the troposphere during the rest of the year (Creamean et al., 2022).

The cloud top temperature histograms in Fig. 11f indicate that a large fraction of the stratiform clouds (Pl + MP clouds) showed top temperatures from -18 to -22°C. A broad distribution with cloud top temperatures from +12°C down to -24°C was observed in the case of PL clouds. The MPC cloud top temperatures covered the range from -12°C to -34°C, with only a few exceptions towards higher temperatures.

The observations in Fig. 11c and f are in good agreement with respect to vertically resolved cloud height and cloud top temperature distributions, derived from multi-year spaceborne lidar and radar observations over the Arctic and presented by Carlsen and David (2022). A strong difference between winter (December to February) and summer (June to August) conditions regarding heterogeneous ice nucleation was found. While in summer, significant ice nucleation started at -10 to -13°C, winter cloud top temperatures had to drop below about -17 to -24°C before ice nucleation sets in. As mentioned, during

the long winter season mineral dust particles prevail as INPs, while during the short summer period biological particles and particles carrying biogenic substances also contribute to ice nucleation.

## 6 MPC conceptual model: an update






The integration of the new dual FOV polarization technique into a state-of-the-art lidar-radar supersite enabled us to obtain a deeper insight into the evolution of Arctic MPCs. The improved monitoring of the ice phase together with the liquid phase, simultaneously and independently of each other, and this with high temporal resolution allowed us to study the different stages of the life cycle of MPCs in large detail. This motivated us to update the conceptual model as it was presented by Morrison et al. (2012).

Briefly summarized, according to the conceptual model, a freshly formed opaque, shallow liquid-water cloud layer leads to strong longwave radiative cooling that initiates the formation of cluster-like updraft and downdraft fields of several kilometers (1-8 km). These larger scale turbulent structures develop even below the opaque cloud layer and influenced the air motion down to the surface, and thus can transport water vapor, CCN and INPs, after sublimation of ice crystals in the virgae, back into the cloud top layer. During updraft periods liquid and ice production preferably takes place (and respective heat release) and keeps the full MPC system alive as long as the synoptic scale humidity conditions allow for liquid water production. Similarly, droplet evaporation and ice crystal sublimation processes (and related cooling of the air) may strengthen the downdraft parts of the circulation pattern.

Based on our MOSAiC lidar and radar observations we can add the following points to the conceptual model:

- (1) CCN activation seems to play an important role throughout the lifetime of a MPC by continuously refilling the small-droplet fraction and stabilizing the droplet size distribution as a whole in this way. This could be concluded from the new dual FOV lidar observations. The CCN reservoir can be estimated from lidar observation whenever cloud-free conditions are given. As in the case of the INP reservoir, the CCN reservoir contains the activated particles as well as the residual activatable CCN. The year-round MOSAiC observations suggest that the CCN reservoir in the free troposphere over the central Arctic was always well fill.
- (2) The MOSAiC lidar observations confirm former studies (Ansmann et al., 2009; de Boer et al., 2011; Westbrook and Illingworth, 2011) that immersion freezing is the most relevant ice nucleation process in stratiform mixed phase clouds. A time-dependent ice nucleation mechanism paired with a large INP reservoir can sustain the frequently observed continuous ice crystal production over many hours (Knopf et al., 2023). Persistent ice nucleation over 2-5 h and 5-10 h was observed in 30% and 13% out of all analyzed 147 MPC layers, respectively. The MOSAiC observations corroborate the hypothesis of a large free tropospheric INP reservoir that was always well filled, i.e., never empty.
- (3) Recycling and entrainment of water vapor, CCNs and INPs from below and entrainment of CCN and INP from above seem to be responsible for refilling of the two reservoirs of CCNs and of INPs. Precipitation can be regarded as the main sink of water vapor, CCNs and INPs. Very clean conditions as frequently observed at Arctic surface stations during the summer half

year, when Artic haze is absent, seem to be only possible in the boundary layer, i.e., in the lowermost 250 m of the troposphere where wet removal by fog and precipitation plays an important role as it was discussed in detail in Ansmann et al. (2023).

- (4) Only in a few cases, ice nucleation was found at temperatures  $> -15^{\circ}\text{C}$  in the free troposphere in the absence of any seeder-feeder impact. This may indicate the dominance of dust particles in ice nucleation processes in the free troposphere. Dust particles are well activatable at temperatures around  $-20^{\circ}\text{C}$  and lower. However, our lidar observations, presented in this article, do not cover the lowermost 500 m of the troposphere, where a direct impact of local aerosols (nordic soil dust, biological material, biogenic substances) on cloud evolution is possible. Low clouds and fog in the Arctic boundary layer with typical depths of less than 250 m occur approximately 50% of the time.
- (5) As a contribution to the discussion whether Arctic MPCs dissipate due to insufficient aerosol or insufficent water vapor supply (Loewe et al., 2017; Sterzinger et al., 2022), the MOSAiC lidar-based aerosol and water-vapor observations in combination with the dense MOSAiC radiosonde humidity profiling suggest that the dissipation of the cloud layers in the free troposphere during MOSAiC was related to a decrease in available water vapor.
  - (6) The case with the strong, distinct perturbation of the cloud evolution over several hours, observed on 21 September 2020, provides an impression to what extent orographically induced perturbations of the main airflow, e.g., by hills and mountains, may be able to temporally or even entirely influence the evolution of stratiform layered clouds.

#### 7 Conclusion/Outlook





We presented the results of the one year MOSAiC observations of Arctic liquid-containing clouds with continuously running lidar and radar instrumentation aboard *Polarstern*, which drifted with pack ice close to the North Pole over many months. For the first time, the full winter half year from October to March was covered with aerosol and cloud observations in the North Pole region. We introduced the new dual FOV polarization lidar technique that allowed, for the first time, a robust and trustworthy monitoring of the liquid phase of stratiform MPC layers. In combination with the lidar-radar retrieval method, a detailed study of the life cycle of mixed phase clouds in terms of the liquid and ice phase microphysical properties was possible.

We discussed two case studies of the temporal evolution of long-lived free tropospheric MPCs. The observations provided new insight into the interplay between the liquid and ice phase of Arctic MPCs. These observations demonstrate that CCN activation is an important process to assure a longevity of a MPC system. In modeling approaches, it is recommended to implement a time-dependent INP parameterization scheme to properly simulate the evolution of long-lasting MPC cloud layers. For the first time, we presented statistical results for the liquid phase of liquid-bearing cloud layers in terms of microphysical properties for an entire year, covering all four seasons of the year.

As an outlook, we will continue with our MPC studies (temporal evolution studies, statistics as a function of aerosol pollution) by means of the new dual FOV technique implemented in lidar-radar supersites. Meanwhile we have long data sets from contrasting observations in Dushanbe, Tajikistan, Limassol, Cyprus, Punta Arenas in southern Chile (Radenz et al., 2021b), and Antarctica (Radenz et al., 2024). We plan long-term observations (2025-2026) in southern New Zealand. A first MPC lidar study was performed by Hofer et al. (2024). These authors analyzed a long-term data set of MPC cloud layers, measured with

lidar over New Zealand, and found a higher ice formation efficiency for clouds which were influenced by continental (Australian) aerosol and a significantly lower ice formation efficiency for clouds which were more influenced by Antarctic/marine aerosols arriving from the Southern Ocean. Our overall goal is to characterize MPCs in a variety of different climate zones and at different environmental conditions to learn more about the aerosol impact to support MPC and climate-change modeling efforts.

## 8 Data availability

Polly lidar observations (level 0 data, measured signals) are in the PollyNet database (Polly, 2024). All the analysis products are available at TROPOS upon request (polly@tropos.de) and at https://doi.pangaea.de/10.1594/PANGAEA.935539 (Ohneiser et al., 2021). Cloud radar data are downloaded from the Cloudnet database at https://doi.org/10.60656/00945b67503743f0 (Engelmann et al., 2023) MOSAiC radiosonde data are available at https://doi.org/10.1594/PANGAEA.928656 (Maturilli et al., 2021) Backward trajectory analysis has been performed by air mass transport computation with the NOAA (National Oceanic and Atmospheric Administration) HYSPLIT (HYbrid Single-Particle Lagrangian Integrated Trajectory) model http://ready. arl.noaa.gov/HYSPLIT\_traj.php (HYSPLIT, 2024). LIRAS-ice products are available from the corresponding author upon request.

## 9 Author contributions

The paper was written and designed by CJ and AA. The data analysis was performed by CJ, KO, MR, HB, JB, and HG. DAK, PS, and UW were involved in the interpretation of the findings. RE, HG, MR, JH, and DA took care of the lidar observations aboard *Polarstern* during MOSAiC. SD was responsible for high-quality MOSAiC *Polarstern* radiosonde launches. All coauthors were actively involved in the extended discussions and the elaboration of the final design of the manuscript.

## 10 Competing interests

Daniel A. Knopf is a member of the editorial board of Atmospheric Chemistry and Physics.

## 760 11 Financial support

765

The Multidisciplinary drifting Observatory for the Study of the Arctic Climate (MOSAiC) program was funded by the German Federal Ministry for Education and Research (BMBF) through financing the Alfred Wegener Institut Helmholtz Zentrum für Polar und Meeresforschung (AWI) and the *Polarstern* expedition PS122 under grant N-2014-H-060\_Dethloff. The lidar analysis on smoke-cirrus interaction was further supported by BMBF funding of the SCiAMO project (MOSAIC-FKZ 03F0915A). The radiosonde program was funded by AWI awards AFMOSAiC-1\_00 and AWI\_PS122\_00, the U.S. Department of Energy Atmospheric Radiation Measurement Program, and the German Weather Service. This project has also received funding from

the European Union's Horizon 2020 research and innovation program ACTRIS-2 Integrating Activities (H2020-INFRAIA-2014 - 2015, grant agreement no. 654109) as well as from the European Union's Horizon Europe Programme under Grant Agreement No. 101137639 (CleanCloud). We gratefully acknowledge the funding by the Deutsche Forschungsgemeinschaft (DFG, German Research Foundation) – project no. 268020496 - TRR 172, within the Transregional Collaborative Research Center "ArctiC Amplification: Climate Relevant Atmospheric and SurfaCe Processes, and Feedback Mechanisms (AC)3". DAK acknowledges support by U.S. Department of Energy's (DOE) Atmospheric System Research (ASR) program, Office of Biological and Environmental Research (OBER) (grant no. DE-SC0021034).

Acknowledgements. Data used in this article were produced as part of the international Multidisciplinary drifting Observatory for the Study of the Arctic Climate (MOSAiC) with the tag MOSAiC20192020 and the Project\_ID: AWI\_PS122\_00. We would like to thank everyone who contributed to the measurements used here (Nixdorf et al., 2021). Radiosonde data were obtained through a partnership between the leading Alfred Wegener Institute, the Atmospheric Radiation Measurement user facility, a U.S. Department of Energy facility managed by the Biological and Environmental Research Program, and the German Weather Service (DWD). We would like to thank the RV *Polarstern* crew for their perfect logistical support during the one-year MOSAiC expedition.

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
