# Peer review of "MOSAiC studies of long-lasting mixed-phase cloud events and analysis of the liquid-phase properties of Arctic clouds"

_EGUsphere, 2025_

## Author Comment (AC1)

Dear reviewer,

thank you for taking the time to carefully read the paper and to provide us with such a long list of recommendations. They are very welcome. We are always very happy and grateful for such an impact.

We considered almost all of the suggestions.

Our answers in blue, and significant changes are in deep blue in the revised manuscript.

**General comments to the manuscript**

In the study titled "Life cycle studies and liquid-phase characterization of Arctic mixed-phase clouds: MOSAiC 2019-2020 results" by C. Jimenez et al., the authors show results of long-term lidar-radar observations onboard the RV Polarstern during the MOSAiC cruise. Radiosonde profiles helped to interpret the data. Four detailed case studies are presented: two of them explain liquid- and ice phase retrieval results, two only liquid retrieval results. Furthermore, statistical results related to free-tropospheric stratiform liquid-containing cloud layers were presented.

**Recommendation:**

I would suggest the manuscript to be published after minor revisions considering the remarks below. The authors should address the following points:

**General/Major comments:**

Title and throughout the text: Is "life cycle studies" the most fitting term? Throughout the study it was not clear to me how the life cycle of the cloud is assessed. Firstly, for the presented statistical analysis the focus is not the life-cycle and thus it is somewhat misleading in the title? Secondly and more generally, I think the term "temporal evolution" is more fitting than life-cycle. You acknowledge on lines 344-346 that from observations at a fixed location (Eulerian perspective) it is hard to perform life cycle analysis (Lagrangian perspective) – but you don't say how you can be sure you really do study the life-cycle as claimed. I suggest referring to the case study analysis as "temporal evolution" unless you can convincingly show that the onset/end of the observation of the cloud over RV Polarstern marks the formation/dissipation of the cloud and not the times the cloud was advected over the observatory.

We removed or changed almost all 'life cycle' statements, throughout the text.

Further remark on title: At the same time, "2019-2020" after "MOSAiC" can be omitted. Furthermore, in Section 5 results of "Pure liquid" clouds are presented. This should be added to the title. How about rephrasing the title to sth. like "Characterization of Arctic liquid-containing free-tropospheric clouds observed during MOSAiC"

We changed the title to cover precisely the contents, i.e., the two parts of the article.

Third remark on title: "liquid-phase characterization" is somewhat misleading: In the first two case studies, liquid- and ice phase are characterized. Why was the ice phase not characterized in the statistical analysis in Section 5? It is strange that case study analysis is done for liquid- and ice-phase but the statistics are not. Remove the ice-phase analysis in case study 1 + 2?

The title now indicates the two parts of the article. The first part deals with long-lasting MPCs and the second part deals with the liquid phase properties of Arctic clouds.

We explain now in the introduction that we only show liquid-phase statistics because an automated data analysis scheme is required to analyze so many cloud cases. And such an automated scheme is not available for the ice phase (for the lidar-radar data analysis), but for the liquid phase (dual FOV lidar data analysis).

Section 4.3 with two more case studies comes as a surprise, as the abstract and the conclusions section mention only 2 case studies. Also, why was only the liquid phase analyzed for these case studies? What is the added value of having four instead of two case studies? – I find the two additional case studies do not add much new content to the manuscript, consider removing them.

We removed the section on June-July MPC case studies (Sect. 4.3). We integrated the 17-18 June observation into the statistics section (Sect. 5) and discuss this case only briefly as an example of the statistical analysis.

It would be good to firstly, mention the limitations of the lidar-based retrievals more clearly (briefly done on line 317): Complete lidar attenuation at optical depth > 2.5-3 leads to underrepresentation of multilayer cloud situations.

We mention now earlier (in Sect. 2.4) that the lidar allows only an accurate cloud optical depth (COD) retrieval up to 2.5. We state also that we only analyzed the lowest cloud layer of multilayer cloud systems with the dual FOV lidar technique. The COD<2.5 constraint has no impact (causes no bias) on the dual FOV lidar products.

**Minor comments:**

Throughout the text many facts are added in brackets – consider removing those or splitting the sentences in two to improve readability.

We reduced the use of brackets as much as possible.

Line 2: Were MPC really only observed in the free troposphere during MOSAiC? – If not, please remove the "free troposphere" here and refer to it later.

We carefully went through the text and mention more often that we analyzed clouds observed at heights above 500 m (and we removed 'free troposphere' statements in this way). According to the MOSAiC article of Peng et a. (2023) the PBL above Polarstern was only exceptionally higher than 500m, but usually clearly below 500 m height (we provide this information now in Sect. 2.4). However, we better leave it open and try to avoid 'free tropospheric cloud' statements. The statistics certainly includes some boundary-layer liquid-containing clouds.

Line 16-17: Is it possible to be more exact than stating "aerosol reservoirs of CCN and INP are well-filled"?

We provide numbers for CCNC and INPC now in addition, in Sect. 4.1.2 (29-31 Dec case study) and 4.2 (21 Sep case study). But the information 'well filled' is already a good statement, to our opinion. The reservoir is simply not empty. There are always sufficient CCN and INPs.

Line 53 – 69: Consider reordering/adding an introductory sentence, so that it becomes clear, that in this paper both, retrievals for liquid phase based on lidar-only observations and ice phase based on lidar-radar observations, are employed.

This now clearly stated in the introduction section. We write:

In the second part of the article, the results of the statistical analysis of the liquid-phase microphysical properties of liquid-containing cloud layers, observed from October 2019 to September 2020, are presented and discussed. The corresponding statistics for the ice phase can unfortunately not be provided. The statistical analysis of the large MOSAiC remote sensing data set requires automated versions of the retrieval procedures. Such an automated version was developed in the case of the dual-FOV lidar data analysis scheme, but could not be realized until now in the case of the lidar-radar retrieval procedure. The lidar-radar data analysis is complex and time consuming and includes a careful selection and setting of input parameters. As a consequence, this retrieval scheme could only be applied to a few case studies.

Line 77: Here you state the focus is on liquid-phase properties. In line 67-68 you state ice phase properties are also retrieved. – So the reader would assume the focus is on both, liquid and ice? – Clarify.

The second part of the introduction is completely rewritten and now it is precisely stated what we present in this paper (in Sect. 4 and 5).

Line 92: Add that the ocean was also studied in depth during MOSAiC.

Done, in Sect. 2.1.

NOTE! Section 2 has an improved structure. We increased the number of subsections from 2.1-2.4 to 2.1-2.6. Furthermore, we introduced a sketch (Fig.1) that explains what we did with the different data sets and what height levels (75 m above cloud base, 250 m below cloud base) are of importance. In this way, the different computations and different retrieval schemes are better visible. All in all, the structure of Sect.2 is less compact and hopefully less confusing for readers not familiar with lidar and radar observations. In addition, we improved Table 1, as requested, but reduced the number of parameters. Only the ones needed in this article are given.

Line 96 - 101: Who is "we"? - It is very uncommon to refer to a group of co-authors as "we" and to focus the literature study on own publications that are not pertinent to the study subject of the manuscript. Consider removing reference to wildfire smoke publications. Consider merging this paragraph with the one on lines 102 – 108 and extend your references to other studies using the MOSAiC atmospheric remote-sensing instrumentation, e.g. https://egusphere.copernicus.org/preprints/2024/egusphere-2024-2193/egusphere-2024-2193.pdf or https://acp.copernicus.org/articles/23/14521/2023/ , https://doi.org/10.1525/elementa.2021.000071 among others.

We rephrased……. and followed the recommendations given in this statement. We removed references, not needed in this article, and added the recent MOSAiC papers as requested.

Line 120: The acronym MOSAiC has been introduced before and does not need to be explained again here.

We removed it.

Line 127: add "profiles of"

Done!

Line 128: "retrievals of" instead of "observations of"

Done!

Line 147: "vertical profiles" instead of "height profiles"

We changed the text in this new subsection 2.3 and considered this remark!

Line 149: Either expand on the "even" by explaining what is special about summer aerosol conditions" or remove it

We rephrased the text in this new subsection 2.3 and considered this remark!

Line 152: I am confused by the wording "reservoir" – Why not call it "proxy for INP concentration"?

We introduced a new paragraph on time-dependent and time-independent INP parameterization (immersion freezing mode) on page 3 of Sect. 1. In this context, the INP reservoir is introduced right in the beginning of the article. The two reservoirs of CCN and INPs are essential to characterize the potential impact of aerosols on droplet and ice crystal nucleation. In the revised version, we use CCN reservoir and INP reservoir throughout the text.

Line 156: replace "our" by "the"

Done!

Line 157: add "troposphere", add a sentence on why altitude ranges below 500m and above 7 km are excluded from the analysis.

We added 'troposphere'.

We stated: To avoid a potentially sensitive bias by the incomplete overlap between the laser beam and the different receiver field of view of the lidar we excluded the dual FOV lidar observations in the near range (< 500 m height) from the further analysis.

 We rephrased and improved the text and state that we looked at all MPC observations at heights above 500 m and found MPCs up to 7 km. There were no MPCs higher up.

Line 161: add "liquid-containing" before cloud layer

Done!

Line 172: move the "well" to the end of the sentence

Done!

Line 175, line 232, line 237 etc: "On" the order of

Improved!

Line 182-183: This sounds confusing: What is the MPC top layer? Is it the liquid-containing layer? - Then you could refer to the base of it as "base of the liquid-containing layer of the MPC" instead of as "cloud base" (here and elsewhere, e.g. line 314)

We changed that throughout the manuscript, to meet the point of the reviewer. We now use 'liquid-containing' or 'liquid-bearing' cloud layer, base of the liquid-containing cloud layer, top of the liquid-containing cloud layer, etc.

Line 184: What are these virga representative for? The evolution of ice properties (e.g. IWC, ice particle effective radius) with increasing distance from the base of the liquid-containing layer is e.g. dependent on the relative humidity. – Expand/clarify.

We rephrased the text and write: According to the airborne in situ MPC observations of Mioche et al. (2017) the ice-phase retrieval products just below the main cloud deck, represent well the ice properties in the lower half of the liquid-bearing cloud layer. In the upper half, the ice crystal properties change much with height as a function of varying ice nucleation rates and the strong increase of the crystal size by water vapor deposition.

We do not mention any impact of relative humidity on the virga structure to avoid another speculative discussion. As long as the virga length (vertical extent below the base of the liquid-containing cloud layer) is 500 m and more, as it was usually the case, we assume that the relative humidity over ice is always around 100% or even a bit higher in the virga zone, so that further growth of ice crystals is small or zero. At the same time, we assume that sublimation of crystals can be ignored at 250 m below cloud base if the virga are longer than 500 m below cloud base. The MOSAiC radiosonde profiles are in line with our assumptions.

Line 210: Are only single-layer stratiform clouds considered or also multi-layer scenarios? Why are clouds > 7 km excluded from the analysis?

As stated in Sect. 2.6, we only analyzed the lowest cloud layer of multi-layer cloud systems. The dual FOV lidar method is very accurate in the case of the first cloud layer, only. Then the depolarization ratios are already 'contaminated' by the first cloud layer so that the measurements in the second layer are biased in a quite undefined way.

Line 218: add "interpolated" to the radiosonde temperature information

Done!

Line 223: Why is a cloud still counted as the same cloud if the cloud-free gap is almost an hour? It seems like a very high allowed gap time. Please clarify.

We extended the discussion in Sect. 2.6. We write: We defined cloud layers as single, individual layers, when they were detected at different heights. In the case of a broken cloud fields (many cloud segments at the same height level), we counted a cloud field as one single cloud system if the detected cloud-free periods lasted for less than an hour. Shupe et al. (2006) counted individual cloud layers in the same way, i.e., cloud layers with gaps of < 1 h in duration were considered to be continuous. If a cloud-free period between subsequent cloud fields exceeded 60 minutes, the next cloud field, crossing Polarstern at that height level, was counted as a new cloud. We assume in this specific cloud length statistics that all cloud segments, separated even by 30-60 minutes, still

developed at the same meteorological and aerosol conditions and, thus, should not be counted as individual, independent cloud layers.

Line 236: What is meant by "time interval of ice nucleation of 60s"?

We rewrote the entire aerosol section (Sect.3) and avoid to mention details of INP parameterizations in which the relevant time interval of ice nucleation is an input parameter. We now give numbers for the reservoir of potential INPs based on the aerosol particle numbers in Fig. 2.

Line 240-241: The sentence "the CCN and INP reservoirs are well-filled" is used 4 times throughout the manuscript. – I still don't understand it. Please only this phrase once and rephrase elsewhere to give readers the chance to understand the meaning once differently expressed.

We reduced the statement of a 'well filled' reservoir, only two statements are left, one in the abstract and one in Sect. 6 (conceptual model update).

The INP reservoir is now introduced in Sect.1. The INP reservoir is needed when the INP parameterization (in models) is a time-dependent approach. All this is now explained in Sect.1.

Line 227 – 242, Fig 1: You previously mention that you consider clouds with tops up to 7 km. Please motivate clearly, why you only show the particle number concentrations at 2 km height instead of at different altitudes.

We improved the figure (now Fig.2), motivated by this comment. Now we show time series of the particle number concentration, n50, for height levels of 1, 3, and 5 km height.

Line 243: Begin what? – Consider removing the phrase.

We removed it.

Line 257: Is the start of the winter-time MPC Dec 30 as stated here or Dec 29 as stated on line 246?

We improved this: 29-31 Dec. in the text and in the figures.

Line 260: clarify if you mean horizontal or vertical wind velocities

Improved: horizontal wind velocities

Line 263: what do you mean by "a few percent of the air mass were advected from 30-60 N"?

We rephased the entire text regarding air mass transport in Sect. 4.1.1.

Now we write: According to the aerosol source identification scheme of \citet{Radenz2021a}, the air mass at 1-3~km height traveled mostly within the European and North American sector at latitudes $>$60°N during the last 10 days before reaching the MOSAiC field site. Arctic haze pollution dominated the aerosol conditions in the lower free troposphere in the central Arctic to that time \citep{Engelmann2021, Ansmann2023}.

Line 264: I suggest adding "likely" in front of "soil material" as soil moisture content also plays an important role in lifting of soil dust that is not considered

We changed the text and soil dust is no longer mentioned.

Line 310: Can you substantiate your hypothesis that riming occurred with the available observations? Also, explain how would riming lead to strong ice production?

We skipped the statements concerning riming here. This would end up in a speculative argumentation and new questions would arise on the potential impact of riming.

We use 'riming' only once and write: The potential impact of riming processes is not discussed here. These processes seem to be unimportant at temperatures $<-20$°C \citep{Waitz2022}.

Line 312: Substantiate your claim of homogeneously distributed ice crystals in the virgae column.

We rephrased the sentence to avoid a discuss on the homogenous distribution of ice crystals from top to bottom of the MPC system. Is not need to be discussed here.

Line 321: add "lidar volume" in front of depol ratio

Done!

Line 321-324: Following your explanation in this paragraph, the low lidar volume depolarization ratios marked in green at the lower end of the virgae are caused by droplets backscattering. – Clarify/expand.

We provide more details to the range of depolarization ratio values in the case of droplet multiple scattering and in the case of ice crystal backscattering including the low values in the lowest part of the virgae where the crystals sublimate and the Rayleigh depolarization value (of 0.01) increasingly dominates the volume depolarization ratio.

Line 332: Consider displaying the cloud radar mean Doppler velocity time-height display to see if you can identify the same virgae structure as well as cycles of up- and downdrafts (superimposed on particle fall velocity) in it. This might substantiate your hypothesis of decreased up- and downdraft strength in the later part of the case study observation as well (lines 355-359).

The Doppler velocity time-height display does not indicate changing updraft and downdraft strengths towards the end of the lifetime of the MPC. So we removed all statements regarding decreasing up and downdraft strengths and write that the decreasing water vapor content of the advected air mass was responsible for the dissolution of the cloud.

Line 336-337: consider discussion of the Sep 21, 2020 case study to its corresponding section. What can we learn from differing horizontal separation of the updrafts in the two considered case studies?

We provide the following text now in Sect 4.1.2, just to provide some information about potential reasons:

Note that during the late summer MPC event, observed on 21 September and discussed in Sect. 4.2}, 40-60~virgae occurred within six hours and point to horizontal separations of updraft zones by only 1.8-2.2~km. Horizontal wind velocities were again around 5~m~s$^{-1}$. The difference between the winter and late summer observations may be related to different meteorological and cloud optical properties which determine the strength of cloud top cooling and thus the up and downdraft characteristics. After formation of the MPC deck the cloud top temperature decreased by 5~K on 30~December 2019 and by 2~K on 21 September 2020.

We think, one needs LES modeling …. to learn more!

Line 340: What is the other measured depolarization ratio?

The ratio of the two depolarization ratios changes with increasing height above the liquid-containing cloud base and depends on the size of the droplets (effective radius) and on the height of the cloud base. In this specific case here, the depolarization ratio for the narrow field of view (shown in Fig.4) is 10 to 20%  lower than the one for the larger field of view. In the ice virga, both depolarization ratios are equal because of the rather strong forward scattering peak of ice crystals so that the receiver field of view does not play any role.

Line 341-343: This was mentioned earlier and can thus be removed.

We removed this statement.

Line 353-354: You attributed the enhanced ice virgae at the beginning of the observation period to potential ice seeding from the cloud above. The strong virgae extends to after when the upper left cloud was not observed anymore (until around 9 UTC). Why?

We improved the discussion in the following way in Sect. 4.1.2: The radar observations in Fig. 4a show a strong virga field above the MPC until 7~UTC so that ice crystals could in principle enter the MPC from above (as ice seeds). The additional ice production may have caused the intensification of ice virga backscattering during the initial phase of the MPC lifetime. However, the top of the liquid-bearing cloud layer increased until 7:00~UTC as well. The increase of the top height is associated with a further decrease of the cloud top temperature and corresponding increase of the ice nucleation efficiency of INPs, resulting in increasingly strong ice crystal nucleation and overall ice production and thus intensification of virga structures. The ice crystal extinction coefficient increased as long as the cloud top height increased. Afterwards, the ice extinction coefficient decreased quickly until 8-9~UTC and then remained almost constant until 23~UTC.

Line 364ff: Please add the definitions of the ice-phase fractions from IWC, LWC, CDNC, and ICNC. – ok, partly shown on line 401, move here at first mention

This is now improved.

Line 38?ff: To me the conclusion that a time-dependent INP activation is central for the longevity to MPC should be added to the abstract.

We included a respective statement in the abstract (at the end of the abstract).

In the discussion of the wintertime case study (Section 4.1.), the feature at 22-23 UTC below 1 km with increased values of several parameters is not mentioned yet and should be discussed.

We now included: An interesting observation was made between 22 and 23~UTC in lowest troposphere at heights below 700~m. The radar and the lidar detected reflection and depolarization features that pointed to the presence of ice crystals. The RHw profile of the radiosonde launched at 23~UTC showed high humidity values close to 100\% and temperatures of $-24$°C. Formation of rather thin water clouds and subsequent immersion freezing and growth of ice crystals as well as strong growth of ice crystals falling into this ice-supersaturated layer from above may have been responsible for the observed ice crystal backscattering features.

Line 418: 88.5°N is "near" the North Pole, not "over" the North Pole. Please correct it.

We changed that!

Line 420: Rephrase "the air mass came from Iceland, Greenland, northern Canada, and even from Alaska" – unclear how the same air mass can come from all of these different directions.

We rephased this part: Similar to the air mass transport on 29-31~December 2020, the free tropospheric air mass, in which the MPC developed, crossed areas mainly north of 60°N (Northern Canada, Iceland region) during the last 10 days before arriving over {\it Polarstern}.

Line 431: Do you have a reference to substantiate your assumption of gravity wave crossing over RV Polarstern?

We do longer mention 'gravity wave'. We write: A strong perturbation of the cloud evolution occurred at 12:00~UTC. This noon event is highlighted in Fig. 6. The reason for the perturbation, which influenced the further development of the MPC system for hours, is unknown and will not be discussed here.

Line 432: In which way does "the gravity wave significantly disturb the development of the liquid and the ice phase of the MPC deck and the interaction between both phase for hours." – In Fig 5, I don't see evidence of a disturbed development, if anything, the development of ice phase seems enhanced (enhanced radar reflectivity) and the lidar volume depol ratio seems to have similar values in the liquid-containing cloud-top layer.

We provide an improved discussion of the 21 September case study. The three height time displays in Fig. 7 (radar reflectivity, lidar extinction, lidar depolarization ratio) unambiguously indicate that the perturbation triggered strong ice formation. The lofting event and associated cooling of air parcels intensified ice nucleation, growth of crystals and caused finally stronger virga structures. Then it gets complicated how the ice formation influenced the properties of the liquid phase. All this is now better and more consistently described, we think.

Line 434-435. +section 454-459: Please list in which products you see perturbations. – I don't see any at the indicated times. Also, this paragraph seems very speculative. Often the word "expected" is mentioned and then it is acknowledged that the expected behavior of variables was not observed. – As the information content is thus limited, I suggest shortening this section considerably.

We shortened the discussion. We now write: The cloud radar observations are shown in Fig.~6b and c. The strong perturbation at 12:00~UTC is not directly visible in the cloud reflectivity and Doppler velocity height-time displays. …

Then we briefly describe what is shown in Fig 6.

Line 446: The term "precipitation fields" sounds not appropriate, the radar reflectivity is very low suggesting that just few ice crystals fell below the main virga features without sublimating, I suggest rephrasing. Please mention if any precipitation was observed by ground-based sensors.

We rephased it. Precipitation did not occur.

Line 450-451: The scale of Fig 5 is too coarse to see the mentioned features.

As mentioned, we changed the text and removed the sentence given in the submitted version in lines 450-451.

Line 460-463: The methodology was previously introduced and can be omitted here.

Done!

Line 466: State why you think riming occurred. – Do you see that in specific variables?

We removed this statement. According to the studies of Waitz et al. (2022) riming does not play a role at temperatures of less than -20°C

Line 472: You mention that "The stable phase in the MPC evolution could not establish before 15:00 UTC." – The ice production from 12 – 15 UTC lasted three hours and thus seems pretty stable to me. Why is this not considered as stable?

The context indicates, to our opinion, that 'stable' means (for us) that some kind of equilibrium, i.e., constant conditions, is reached. And these constant conditions are not found before 15 UTC.

Line 479-480: Explain why the alternative hypothesis is not convincing.

We skipped this sentence. We want to simply emphasize that the observations are consistent with CCN activation. Whether this CCN activation is a result of changing aerosol conditions or of other reasons is not needed to be known in this context.

Line 481-482: It is stated that ice crystal effective radius was 50 microns during the stable phase of the MPC. In Fig 6, it looks like as if this was the case for the entire observation period. – Clarify.

We agree and mentioned that now.

Line 485: 2x "alpha_liq" used

We improved it.

Line 490: Not quite true, IWC peaked again at 15 UTC.

We improved the discussion, and discuss even the 15 UTC event.

Line 488-489: description of LWP time series is incomplete (10-15 UTC is missing). Why is there no LWP from 11-12 UTC in Fig7 Panel c).

We provide the data now. We checked all available MOSAiC radiometer measurements. All show the same time series for LWP. Now we include the missing data (from 11-12 UTC) and discuss the LWP observation , including the 'strange' part of the time series from 11-12 UTC. The reason for the monotonic increase of the LWP data from 11-12 UTC is most probably the development of another cloud layer at about 1 km height (indicated by the 11 UTC radiosonde). And this liquid-containing cloud layer obviously vanished, when the strong perturbation at 12 UTC occurred. The reason for dissolution of the lower cloud layer is probably ice seeding from above and rapid conversion into ice.

Line 504: does the "most" refer to the entire MOSAiC observation period or to June and July?

The section with the June and July case studies is removed. The 17-18 June 2020 observations are now part of the statistics section and are only very briefly discussed, as an example how we used all the cloud observations and information to end up with a year-round statistics.

Line 517: add a verb to the sentence

Improved!

Line 521-22: at which times do you expect seeding to play a role (not clear to me in Fig. 8 as most pronounced virgae are mostly not at the same time as lower-liquid-containing cloud layers

As mentioned, the section on June-July observations is now removed.

Line 553-554: The sentence is unclear.

We removed the respective sentence. Is not needed.

Line 555: What were the criteria for the selection of the subset?

The subset consists of 360 h of liquid-containing cloud observations. Note that we analyzed 147 cloud cases and not 94.

Regarding the criteria we write in the revised version in Sect. 5 (Statistics section): The observations during these 360 hours fulfilled all signal quality criteria, stated in Sect. 2.6 and the measurement conditions were perfect for dual FOV lidar applications, i.e., fog and low clouds were absent and the analyzed cloud layers showed well defined, sharp cloud base structures.

And in Sect. 2.6 we write: The quality assurance procedure includes checks of the inter-channel constants between all four channels used to determine the two volume depolarisation ratios. Here, long data sets with clouds and cloud-free conditions are used to check the long-term stability of the counting efficiencies of the polarization sensitive channels. It was also checked that none of the lidar signal counts (in each channel) reached the saturation level of the detectors during cloud events.

Line 561: Why are cloud layers observed for < 20min not considered?

We removed the statements with the 20-minute limit. We used all analyzed 7-minute profile averages out of the 360 hours as long as they fulfilled the quality criteria. At the end we had 3070 of these 7-minute profiles.

Line 565-566: How do the statistics of your analysis compare to these values?

First of all, Fig. 10 now includes 147 (and not only 94) cloud layers and also clouds with temporal length less than one hour.

Regarding the comparison with Shupe et al. (2006) we found and write:

In about 50\% out of the 147 cloud events, we observed clouds layers … with temporal lengths of $<2$~h. In about 30\% and 13\% out of all analyzed cases, continuous cloud observation length was 2-5~h and 5-10~h, respectively. Observations that lasted for more than 10~h contributed to 7\% to the total number of 147 selected MOSAiC cloud events. Shupe et al. (2006) analyzed a one-year cloud data set of 284 identified Arctic liquid-bearing clouds measured between October 1997 and September 1998. They found cloud duration lengths of <2~h, 2-5~h and 5-10~h, in 25\%, 20\%, and 20\%, out of all cases.

Line 574: I don't think the pure liquid layers refer only to clouds before ice nucleation sets in: In Fig 11 you show that quite a large fraction of PL layers have CCT of > 0C – so there won't be ice formation setting in. rephrase.

This is improved now.

Line 665: repeat the height range for "low cloud layers" here

We now provide this information: However, our lidar observations, presented in this article, do not cover the lowermost 500~m of the troposphere, where a direct impact of local aerosols (nordic soil dust, biological material, biogenic substances) on cloud evolution is possible.

**Comments on Tables:**

Table 1: Very good that a table regarding uncertainties is included. Please expand by adding two more columns: One indicating if the parameter is lidar-derived or lidar-radar derived and one more adding a reference in which the uncertainty is derived.

Done!

**Comments on Figures:**

Consider using logarithmic scale units for displaying radar reflectivity instead of linear units as commonly done in order to allow visual comparison of reflectivity with other manuscripts focusing on detailed case study analysis.

Done!

Fig. 1: Please explain the reason for the data gaps in the caption.

Done!

Fig. 2: Shorten the caption by removing sentences giving an analysis of the figure (virga is formed etc). Also, it is mentioned that the black vertical lines in b refer to the radiosonde launches at 5 and 17 UTC on Dec 31 etc. In panel b), the vertical black lines are at 0, 6, 12, and 18 UTC though and I count five (instead of four mentioned) black vertical lines. – Correct.

We worked on the caption to consider the points.

Also, in the description of Fig.2, pls comment on the cause of the layers of enhanced signal strength between roughly 1.5-2km altitude and 0-3 UTC.

We discuss now the clouds occurring at 1.5-2 km between 0-3 UTC.

Fig 2, 3 etc: Consider rephrasing "life-cycle" to "temporal evolution" unless you can prove that the onset/end of the observation of the cloud over RV Polarstern marks the formation/dissipation of the cloud and not the times the cloud was advected over the observatory.

Done!

Fig 3: In panel b,c add "lidar" to the title to make it coherent with panel a where you state the instrument name (radar)

Done!

Fig 4: add a horizontal line at 250m below liquid-containing layer base as well as 75m above it to show at which altitudes the values presented in Fig 3 are from.

We added the 250 m line only. We do not like to present a line at 75 m above cloud base in Fig.4. As mentioned, we have a new Fig.1 (a sketch) that shows the different height levels for which we calculated all the different products.

Fig5. +line 452: In the caption refer to Panel c) as "Mean Doppler Velocity" as "vertical velocity" could be mistaken as "vertical air velocity". Shorten the caption by removing the last sentence ("The orange regions may indicate upwind areas when taking permanent ice crystal sedimentation into account.") since it is an interpretation of the figure which belongs to the main text.

Done!

Fig.7: "and produced strong ice virgae and triggered strong ice production." is discussion and should thus be avoided in the caption. What happens at 15 UTC? IWC as well as IWP show a peak and should also be discussed.

We changed the caption text and also discuss the 15 UTC peak now.

Fig 8: shorten the caption by removing "All cloud layers show a blue color at cloud base (not always visible) in panel b, in an unambiguous sign for liquid-dominated cloud layers so that ice is produced by immersion freezing. The strong increase of the depolarization ratio with height (from blue to light greem yellow or even red) is caused by multiple light scattering by the water droplets."

Done!

Fig. 11: Panel e) should have CDNC as x-axis label.

Improved!

---

## Author Comment (AC2)

Dear reviewer,

thank you for careful reading and spending many hours to provide us with a long list of valuable suggestions and recommendations that helped a lot to improve the manuscript significantly.

We considered almost all points.

Our answers in blue in this reply letter. Significant changes are in deep blue in the revised manuscript.

**General comments:**

This paper summarises the new information added to the body of knowledge regarding mixed-phase clouds (MPCs) by analysis of the data collected from the research vessel *Polarstern*. The paper presents four case studies and a statistical analysis of all observations in support of some generalised conclusions about the behaviour and formation of MPCs. This is a worthwhile expansion of the current scientific understanding of MPCs, and my recommendation is for publication after the minor issues outlined below have been addressed.

**Specific comments:**

The main issue that stands out to me is the statement in lines 670–671 that 'The decreasing moisture content of an air mass, rather than empty CCN and INP reservoirs, is probably the reason for the dissolution of stratiform cloud layers in most cases'. While you do clearly establish that in your case studies you observed water droplets nucleating on a sufficient supply of background aerosol, that does not by itself conclusively answer the question of whether most MPCs dissipate because they exhaust their supply of water or because they exhaust their supply of aerosol (c.f. Sterzinger et al. [no affiliation] 2022, 'Do Arctic mixed-phase clouds sometimes dissipate due to insufficient aerosol?' or Loewe et al. [no affiliation] 2017, 'Modelling micro-and macrophysical contributors to the dissipation of an Arctic mixed-phase cloud during the Arctic Summer Cloud Ocean Study (ASCOS)'). Both are physically possible causes. If your argument is that throughout the full year of MOSAiC observations you never saw evidence of cloud dissipation as a result of low aerosol (and thus low CCN and INP), this point should be made more explicitly and in more detail.

We observed so many cloud layers and checked the water vapor conditions when the end of an extended cloud layer crossed the Polarstern, and we always found in the Raman lidar water vapor profile observations (possible during the winter half year) and in the year-round radiosonde humidity profile observations that the cloud layer dissolved because the humidity decreased in the air mass. We never had a case where the humidity remained at high level but the cloud dissolved.

However, we concentrated on the free troposphere, and the aerosol content in the free troposphere is widely linked to the aerosol conditions over the continents surrounding the Arctic via long-range transport. The situation may be very different for boundary layer clouds. Here, efficient washout processes may persistently remove aerosol particles from the boundary layer so that situations may occur where a cloud cannot form because the aerosol content is too low.

We mention our impression (dissipation of the cloud layers by decreasing water vapor in the air mass) only briefly in Section 6 (MPC conceptual model: an update).

I agree with RC1 (no known affiliation) that 'temporal evolution' would be preferable to 'lifecycle', especially in the context of observations taken from a drifting vessel under

advected cloud. Separately, referring to the cloud as 'alive' (e.g. line 35: 'keeps the MPC top layer alive, frequently for hours') is idiomatic and unclear. A description of the cloud processes would always be preferable to an analogy to biological life, e.g. 'The steady resupply of water droplets allows the MPC supercooled liquid top layer to persist despite the continuously forming ice below, frequently for hours'.

This sentence 'The steady resupply of water droplets…' is now included in the second paragraph of the introduction.

The existence of the gravity wave in the September case also seems to me inconclusive. In the first mention on line 249 it is understood to be the 'best guess' theoretical explanation for the upward motion: 'Gravity waves may have caused the perturbations.' Similarly line 431: 'probably the result of a gravity wave', and line 669: ' probably by gravity wave activity'. However, the caption for Figure 7 states definitively: 'Gravity waves crossed Polarstern between 12:00 and 14:00 UTC'. What evidence is there for this?

We removed all speculations about the reason for the perturbation. We leave it simply open. We no longer use 'gravity wave' in the manuscript.

Lines 8–9: 'We discuss two long-lasting Arctic MPC cases (one mid winter case and one late summer case) observed close to the North Pole in December 2019 and in September 2020.' This should be edited to include mention of the 18 June and 25–28 July cases.

We removed Section 4.3 (June-July 2020 MPC observations).  Now, there is no discussion of additional two cases. However, we use the 17-18 June observations as an example for the statistical analysis. Therefore, the 17-18 June case is now part of the statistics section (Sect. 5). The discussion is kept short of 17-18 June observations is kept short.

Line 40: define 'fall strikes'.

We mean: fall streaks, and improved it.

Lines 98–101: Remove the sentence 'We even illuminated a potential role of stratospheric wildfire smoke on polar ozone depletion (Ohneiser et al., 2021; Ansmann et al., 2022) and the relationship between vertically integrated tropospheric water vapor and the downward, broadband thermal-infrared irradiance at the ground during the MOSAiC winter half year (Seidel et al., 2024).' These achievements are not directly relevant to the focus of this paper.

Done!

Lines 137–137: 'The basic lidar data analysis applied to obtain the geometrical and optical properties (backscatter, extinction, linear depolarization ratio) is outlined in Baars et al. and Hofer et al. (2017).' The placement of this parenthetical list suggests that all of these are both geometrical and optical properties. 'Geometrical' also does not occur in either of the sources cited, so the meaning is not immediately clear. I would suggest changing to e.g. 'The basic lidar data analysis applied to obtain the geometrical (cloud base and cloud top heights) and optical (backscatter, extinction, linear depolarization ratio) properties is outlined in Baars et al. and Hofer et al. (2017).'

We used this description now in Sect. 2.2. Note, Section 2 now has more subsections, Sect. 2.1 to 2.6 instead of 2.1 to 2.4 before. Each data analysis approach has its own subsection. The structure of this complex and busy section is now better visible, we think.

Lines 150–152: 'The particle number concentration n50, considering all particles with radius >50 nm, is used as a proxy for the CCN concentrations, and n250, considering the large particle fraction (particles with radius >250 nm), is used to indicate the reservoir of INPs.' Why is one a proxy and the other a reservoir? 'Proxy for the INP concentrations' would make more sense here, especially if you explained immediately thereafter that you are treating 1% of the n250 as ice-nucleating dust.

We included a new paragraph in Sect. 1 on time-dependent and time-independent parameterization of immersion freezing, and in this context, we introduce the 'INP reservoir'.

We now use the wording 'reservoir' throughout the manuscript.

Note, Fig.1 of the submitted manuscript is now Fig.2 in the revised manuscript. In Fig. 2, we now show n50 for the height levels of 1 km, 3 km, and 5 km. In the old Fig. 1 we showed n50 for the height level of 2 km, only. This improvement was indirectly requested by the other reviewer.

Table 1: It would be very helpful to include a column for the abbreviations for each aerosol and cloud property e.g. $R_{e,liq}$ for cloud droplet effective radius.

We improved Table 1 accordingly. Note, Table 1 contains much more information now. On the other hand, we reduced the number of quantities. Listed are only parameters used in this article.

Lines 179–180: 'Since lidar observation of pure ice crystal backscattering is only available for the virga zones' – explain why in physical terms.

This is now explained in detail in Sect. 2.5. The main reason is that cloud droplets do not fall out of the liquid-bearing cloud layer, in contrast to ice crystals. Droplets are too small.

Lines 182–183: 'the cloud base of the MPC top layer' – unclear phrasing. I would suggest changing this to 'base of the liquid-dominated cloud layer of the MPC' as in line 170 and keeping this phrasing consistent throughout the text. A simple schematic diagram might also be helpful to the reader.

We followed both recommendations! We introduced a new Fig.1 (sketch). This will help to better understand our research strategy by combining lidar-radar observations at 250 m below the base of the liquid-containing cloud layer and dual FOV lidar observations 75 m above cloud base.

Lines 183–184 and 199–200: 'these virga observations are well representative for the entire MPC height range, including the liquid-dominated cloud top layer'; 'we use the virga IWC value at the top of the virga zone to be representative for the entire liquid-dominated cloud top layer as well' – justify this briefly in-text as well as giving the Mioche et al. (2017) citation.

We try to explain that a bit better and write in Sect. 2.5: According to the airborne in situ MPC observations of \citet{Mioche2017}, the ice-phase retrieval products just below the main cloud deck, represent well the ice properties in the lower half of the liquid-bearing cloud layer. In the upper half, the ice crystal properties change much with height as function of the ice nucleation rate and growth of the crystals by water vapor deposition.

Line 210: 'After applying several quality assurance procedures to the lidar observations' – detail these.

We provide detailed information now in Sect. 2.6: The quality assurance procedure includes checks of the inter-channel constants between all four channels used to determine the two volume depolarisation ratios. Here, long data sets with clouds and cloud-free conditions are used to check the long-term stability of the counting efficiencies of the polarization sensitive channels. It was also checked that none of the lidar signal counts (in each channel) reached the saturation level of the detectors during cloud events.

Lines 222–223: 'we counted a cloud field as one single cloud system if the detected cloud-free periods lasted for less than an hour' – an hour seems like a very long gap. Explain the selection of this threshold.

We now write: In the case of a broken cloud fields (many cloud segments at the same height level), we counted a cloud field as one single cloud system if the detected cloud-free periods lasted for less than an hour. \citet{Shupe2006} counted individual cloud layers in the same way, i.e., cloud layers with gaps of $< 1$~h in duration were considered to be continuous}. If a cloud-free period between subsequent cloud fields exceeded 60 minutes, the next cloud field, crossing {\it Polarstern } at that height level, was counted as a new cloud. We assume in this specific cloud length statistics that all cloud segments, separated even by 30-60~minutes, still developed at the same meteorological and aerosol conditions and, thus, should not be counted as individual, independent cloud layers.

Lines 229–230: 'The retrieval of these particle number concentrations are explained in detail in Ansmann et al. (2023)' – it would still be good to have a sentence or two briefly outlining the methods here.

In the new Sect. 2.3 (POLIPHON…) we explain the method in more detail now.

Line 240: 'At these high temperatures, mineral dust particles are ice-inactive' – a citation would be good here.

We rewrote the entire Sect. 3 (MOSAiC free tropospheric aerosol conditions). The sentence above is removed.

Lines 296–297: 'Increasing cooling of the MPC top layer also leads to an increase of available INPs.' By what mechanism?

Yes, this statement is misleading.

In Sect. 4.1.1, now we write: Increasing cooling of the MPC top layer also leads to an increase of the ice nucleation efficiency of activatable INPs \citep{Demott2015, Kanji2017, Wex2019}.

That must be sufficient here. It has been shown in the lab, that the efficiency roughly increases by a factor of 10 when the temperatures decrease by 5 Kfor temperatures from -20 to -30°C.

Lines 281–282: 'The longevity of the MPC deck is, to our opinion, the result of the continuous production of liquid water, especially of the formation of new droplets' – the water itself is not being produced. Rephrase to something like 'The longevity of the MPC deck is, in our opinion, the result of the continuous nucleation of liquid water to form new droplets'.

*Improved!*

Line 307: 'seeder-feeder effects' is used on this line for the first time but not defined in physical terms until lines 349–351; move that definition to accompany this first occurrence.

*We changed that accordingly.*

Line 364: 'ice-phase fraction' is used on this line for the first time but not defined in physical terms until line 401; move that definition to accompany this first occurrence.

*Improved!*

Line 379: 'permanently' is an odd word here; rephrase. 'Persistently' would be one alternative.

*Improved!*

Line 387: Explain in greater detail the significance of the time-dependent ice nucleation mechanism in the model and how this relates to your results. This is interesting but seems as though it is mentioned almost in passing.

*This is done already in Sect. 1. A new paragraph is given on this in the Introduction.*

Line 391: 'the number of ice-nucleating particles (INPs) available for ice formation, termed INP reservoir' – move this definition earlier, to lines 150–152 when you first introduce the term.

*Also the 'reservoir' is now introduced in Sect. 1.*

Line 419–420: 'As on 30-31 December 2019, the air mass came from Iceland, Greenland, northern Canada, and even from Alaska.' I suggest rephrasing to something like 'the air mass contained aerosols from …', and clarifying that this was known through the Radenz air mass attribution scheme (assuming that was how) and explaining briefly.

*We rephrased it. As in Sect. 4.1.1 (29-31 Dec case) we avoid to mention sources of aerosols. We only know the transport ways. But this now better described in Sect 4.1.1 and 4.2.*

Line 455: as on line 379, 'permanently' is not the right word here.

*We rephrased this part.*

Lines 456–459: The phenomenon you describe (vertical motions at 12:00 and 12:40) is frankly still not visible to me in Figure 5c despite the adjustments.

*Yes. So, we improved the text We mention that the perturbation is not visible in the radar data. Only the triggered ice formation and intensification of the virga structures is visible.*

Line 479–480: 'The alternative hypothesis that changes in the cloud properties are simply the result of changing aerosol conditions is not convincing' – explain why not.

We skipped this sentence. We wanted to simply emphasize that the observations are consistent with CCN activation. Whether this CCN activation is a result of changing aerosol conditions or of other reasons is not needed to be known in this context.

Lines 528–529: 'Mineral dust particles were probably responsible for strong ice nucleation in the air mass above 6 km height' – what is your reason for assuming this?

We removed the entire Sect. 4.3 (June-July 2020 MPC observations). So, no explanation is needed anymore.

Lines 539–540: 'Mineral dust is the most favorable INP type at these low temperatures' – c.f. line 240, a citation would be good here.

We did that somewhere else in the article. As mentioned, Sect. 4.3 is removed.

Lines 558–559: 'A careful data quality check with special focus on properly aligned dual FOV receiver characteristics was applied to all of the selected cloud events' – again, outline what this check consisted of. An appendix detailing what all of these checks were (c.f. line 210) would be very useful as supplementary material.

Note that we analyzed 147 cloud cases and not 94. The 147 cloud cases covered the 360 cloud hours.

Regarding the criteria we write in the revised version in Sect. 5 (statistics section): The observations during these 360 hours fulfilled all signal quality criteria, stated in Sect. 2.6 and the measurement conditions were perfect for dual FOV lidar applications, i.e., fog and low clouds were absent and the analyzed cloud layers showed well defined, sharp cloud base structures.

And in Sect. 2.6 we write: The quality assurance procedure includes checks of the inter-channel constants between all four channels used to determine the two volume depolarisation ratios. Here, long data sets with clouds and cloud-free conditions are used to check the long-term stability of the counting efficiencies of the polarization sensitive channels. It was also checked that none of the lidar signal counts (in each channel) reached the saturation level of the detectors during cloud events.

We do not like to introduce a supplementary part.

Line 579: 'The histograms of the PL cloud properties in Fig. 11a, b,d, and e are slightly and partly even considerably broader' – this phrasing is not at all clear. Rephrase to something like 'The histograms of the PL cloud properties in Fig. 11a, b, d, and e are all at least slightly broader, and some considerably broader, than the respective frequency-of-occurrence distributions for MPCs'.

We improved that accordingly.

Line 679: 'We presented two case studies' – rephrase to four case studies, unless you have decided to omit the June and July cases.

We removed the Section 4.3 with the two additional case studies.

Lines 685-686: 'The measurements further provided the impression that the CCN and INP reservoirs were always well filled, i.e., never depleted upon ice crystal formation.' This statement only makes sense if you are treating CCN as synonymous with INP – rephrase.

We rephrased it. Note, that we now give numbers for the reservoirs in the sections for the different cases (29-31 Dec, 21 Sep).

Lines 690–691: 'A first MPC lidar study was performed by Hofer et al. (2024)' – add a sentence or two with more detail on this.

Done!

**Technical corrections**

The use of 'rather' throughout (e.g. line 92, 'A rather detailed monitoring of the atmosphere…') sounds informal and could in all occurrences be omitted without changing the meaning of the sentence.

Improved!

'Year-around' and 'year around' are nonstandard and should be replaced with 'year-round' in all cases.

Improved!

Figure 2: 'indicate times with now useful lidar observations' should read 'indicate times with non-useful lidar observations'.

Improved!

Figure 10: 'as a function of the time period, needed by the cloud field to cross the Polarstern' should read 'as a function of the time period needed by the cloud field to cross the Polarstern'.

Improved!

Figure 11: 'The black histogram lines s are based on 3070 cloud data sets' should read 'The black histogram lines are based on 3070 cloud data sets'. Because all the histograms are outlined in black, consider rephrasing as 'The thick black histogram lines are based on 3070 cloud data sets' to differentiate that set of lines.

Improved

Line 50: 'recognozed as the mian' should read 'recognized as the main'.

Improved!

Line 80: 'an introduction in the MOSAiC Polarstern route' should read 'an introduction to the MOSAiC Polarstern route'.

Improved!

Line 113: 'cloud micropyhsical properties' should read 'cloud microphysical properties'.

Improved!

Line 157 and onwards: 'recently introduced' and 'new' are already established as a property of the dual FOV polarization lidar method. This does not need to be repeated on subsequent occurrences.

Improved!

Lines 162–163: 'The multiple scattering effect is a strong function of the number concentration of cloud droplets, their size, as well as of the receiver FOV of the lidar' should read 'The multiple scattering effect is a strong function of the number concentration of cloud droplets and of their size, as well as of the receiver FOV of the lidar'.

Improved!

Lines 275–276: 'The strongest temperatures decrease' should read 'the strongest temperature decrease'.

Improved!

Line 307: 'These crystals may have influence the MPC evolution' should read 'These crystals may have influenced the MPC evolution'.

Improved!

Line 302: 'Microphyscial properties' should read 'Microphysical properties'.

We changed all 'microphyscial' to 'microphysical'.

Line 315: 'clout top layer' should read 'cloud top layer'.

Improved!

Line 337: 'the retrieval products for both, the liquid and the ice phase' should read 'the retrieval products for both the liquid and the ice phase'.

Improved!

Line 350: 'on the expense of liquid water droplets' should read 'at the expense of liquid water droplets'.

Improved!

Line 479: 'taking strong CCN activation into considerations' should read 'taking strong CCN activation into consideration'.

Improved!

Line 494–495: 'covered the sky above Polarstern in 50-55% of the time' should read 'covered the sky above Polarstern 50-55% of the time'.

We removed Section 4.3.

Line 603: 'ice cyrstal growth' should read 'ice crystal growth'.

We improved all 'cyrstal' to 'crystal'.

Line 611: 'Most PL clouds layers' should read 'Most PL cloud layers'.

We improved all 'clouds layers' to ' cloud layers'.

Line 634: 'before ice nucleation sets' should read 'before ice nucleation sets in' or 'before ice nucleation starts'.

Improved!

Lines 665–666: 'Low clouds layers occurr approximately 50% of the time' should read 'Low cloud layers occur approximately 50% of the time'.

Improved!

Line 666: 'the aerosol emitted in the Arctic..' should read 'the aerosol emitted in the Arctic.'

We changed the text.

Lines 681–682: 'These observations demonstrate, that CCN activation is an important process to assure a longevity of an MPC deck' should read 'These observations demonstrate that CCN activation is an important process to assure the longevity of an MPC deck'.

Improved accordingly.

Line 705: 'the elaboration of the final design of the manuscript' should read 'the elaboration of the final design of the manuscript.'

Improved!

Line 707: 'a member of the editorial board of Atmospheric Chemistry and Physics' should read 'a member of the editorial board of Atmospheric Chemistry and Physics.'

Improved!

Lines 774–779: The citation for Baars et al. 2016, 'An overview of the first decade of PollyNET: an emerging network of automated Raman-polarization lidars for continuous aerosol profiling' is incomplete.

Improved!